# A fast heuristic to optimize time-space tradeoff for large models

**Akifumi Imanishi** *
Preferred Networks
imanishi@preferred.jp

**Zijian Xu** *
Preferred Networks
joe@preferred.jp

**Masayuki Takagi**
Preferred Networks
mtakagi@preferred.jp

**Sixue Wang**
Preferred Networks
cecilwang@preferred.jp

**Emilio Castillo**
Preferred Networks
ecastill@preferred.jp

## Abstract

Training large-scale neural networks is heavily constrained by GPU memory. In order to circumvent this limitation, gradient checkpointing, or recomputation is a powerful technique. There is active research in this area with methods such as Checkmake [19] or Moccasin [3]. However, both Checkmate and Moccasin rely on mixed integer linear programming or constraint programming, resulting in limited scalability due to their exponentially large search space.

This paper proposes a novel algorithm for recomputation (FastSA) based on a simulated annealing heuristic that achieves comparable or even better solutions than state-of-the-art alternatives. FastSA can optimize computational graphs with thousands of nodes within 3 to 30 seconds, several orders of magnitude faster than current solutions.

We applied FastSA to PyTorch models and verified its effectiveness through popular large vision and text models, including recent language models with the transformer architecture. The results demonstrate significant memory reductions by 73% with extra 18% computational overheads on average. Our experiments demonstrate the practicality and effectiveness of our recomputation algorithm, further highlighting its potential for wide application in various deep learning domains.

## 1 Introduction

The memory requirements for deep neural networks continue to grow together with model complexity. For example, training a state-of-the-art Large Language Model (LLM) such as LLaMA [40] with 65 billion parameters needs 1024 NVIDIA A100 GPU devices for 21 days and different techniques to overcome the immense memory consumption of the process. While the model parameters may fit in the device memory, the results of several operations of the forward pass are saved and remain in the device memory to compute the gradients during the backpropagation step. This limits the size of trainable models. One of the most widely used solutions is to split the model parameters across multiple devices into vertical or horizontal ways [34, 26] in what is called Pipeline or Model parallelism. However, these approaches may have the following problems. (1) It may be difficult to split the model equally across the workers when the model architecture is complex. (2) We need to modify the model for the parallelism. (3) Hiding communication behind computation needs further tuning. Heavy inter-device communication may result in a GPU utilization of only 5% of the hardware peak[30].

---

*Equal contribution

37th Conference on Neural Information Processing Systems (NeurIPS 2023).

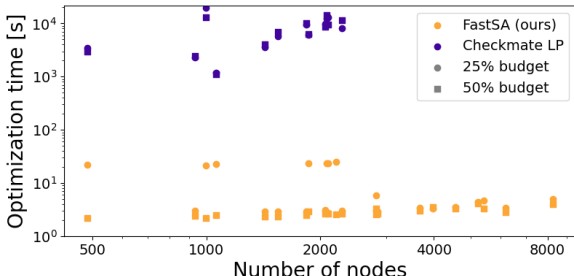

Figure 1: Optimization time for all 23 models used in our experiments with different budgets. Checkmate LP could not find a feasible solution within 6 hours for models with more than 2300 nodes. For most instances, FastSA could found recomputation plan around 3 seconds.

Numerous techniques exist to lessen memory requirements while training deep neural networks. Gradient checkpointing, or recomputation, is a widely-known approach that recomputes some of the activations during the backward pass, rather than saving them in memory, consequently reducing memory requirements. Recomputation plans can be identified by the user or done heuristically. Originally, division-based methods were used for automatic recomputation, where the forward computation graph was split into stages, and only tensors active across stages were stored [7, 21, 20]. However, these methods are less effective for complex large networks that are far from sequential networks. Checkmate is one of the state-of-the-art methods in automatic recomputation. It can be applied to any computational graph, considering the costs of operators and the sizes of values. However, Checkmate requires to solve a large mixed integer linear programming problem, where the search space is exponential to the square of the size of the computational graph, thus requires immense computational time and a large RAM resource. Moreover, Checkmate cannot efficiently reduce the memory usage if the initial computation order is not memory efficient enough (discussed in Appendix A.1). A recent work, Moccasin [3], uses constraint programming to build a set of restrictions and introduces a hyperparameter that limits the amount of times a variable can be recomputed. Although the search space remains exponential like Checkmate, this hyperparameter lessens the number of integer variables from quadratic to linear with respect to the graph size, resulting in a faster execution.

This paper proposes a novel technique to recompute the activations that need to be saved for the backpropagation pass by using a fast heuristic algorithm that operates on the joint computational graph of the forward and backward passes and determines which operations to be recomputed to reduce the amount of used temporal memory. We formalize the problem of recomputation as finding a candidate sequence of operators that minimizes an objective function. We apply a simulated annealing algorithm where each step can be an operator addition or deletion from the computational sequence, or altering the order in which operators are sequentially applied, while maintaining the dependencies between them. The performance of the simulated annealing is significantly improved by using a segment tree data structure [9] which allows to lazily evaluate the memory usage of each candidate sequence, enabling nearly 1 million mutations on the candidate sequence per second.

The main contributions of our paper are:

- We present a novel approach to recompute intermediate results in computational graphs using a heuristic algorithm based on simulated annealing that works for any computational graph.

- We further improve the algorithm's performance using a data structure that allows efficient computation of peak memory usage when the sequence of operators is changed. This can optimize the recomputation within 3 to 30 seconds even for large computational graphs.

- We evaluate our proposal using a representative set of models obtained from Hugging Face, including vision models, text models, and recent language models such as LLaMa [40]. We show reductions in memory consumption by 73% with an overhead of extra 18% computational cost on average.

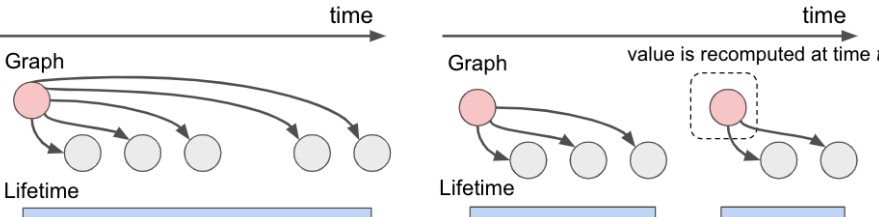

Figure 2: Lifetime of recomputed values. The left side shows the full lifetime of a value without recomputation, while the right side demonstrates that recomputation results in the value having two distinct lifetimes.

## 2 Fast Simulated Annealing for Recomputation

### 2.1 Problem Description

To apply the Simulated Annealing optimization strategy to the recomputation problem, we present a series of formal concepts that will be used throughout the algorithm description.

**Computational Graph and Sequence**   We consider computational graphs in which each node represents an operator. A computational graph consists of a finite set of nodes $\mathcal{N}$ and a finite set of values $\mathcal{V}$. Each node takes some values as inputs and produces some values as outputs; therefore, we introduce notations $\text{input}(n) \subseteq \mathcal{V}$ and $\text{output}(n) \subseteq \mathcal{V}$ for the values associated to a node $n \in \mathcal{N}$. In most cases, the numbers of $\text{input}(n)$ and $\text{output}(n)$ are small. The computational graph is often visualized as a directed acyclic graph (DAG) with directed edges from $n_i \in \mathcal{N}$ to $n_j \in \mathcal{N}$ if an output of $n_i$ is used as an input of $n_j$, i.e., $\text{output}(n_i) \cap \text{input}(n_j) \neq \emptyset$. A value that is not the output of any node is a model input, which often represents the arguments passed to the model and the model parameters. We also have model outputs, the target of the computation. Both nodes and values are weighted: the computational cost of node $n$ is represented by $c(n) \geq 0$, and the size of value $v$ is represented by $s(v) \geq 0$.

Rather than performing the optimization over the DAG, the proposed method is easier to understand if the computation is treated as a sequence of nodes $(n_1, \ldots, n_T)$, which means that $n_t$ is operated at integer time $t$. We say that a sequence of nodes has a valid dependency if we can execute the operators in the order of the sequence so that all model outputs are correctly computed. Formally, $\forall i. \forall v \in \text{input}(n_i). \exists j < i$ such that $v \in \text{output}(n_j)$ or $v$ is a model input. It is worth noting that node duplication is allowed in the above node sequence $(n_1, ..., n_T)$, and the recomputation of a value can be represented by multiple appearances of the same node.

**Memory usage**   In order to optimize memory usage, we must keep track of when values are computed and used as inputs. Specifically, we must keep value $v$ in memory from the time it is computed as an output until the last time it is used as an input. However, when $v$ is recomputed at time $t$, we can temporarily free $v$ from memory if it is not used until $t$. When recomputation is considered, the lifetime of $v$ can be represented as a set of independent time intervals, as shown in Figure 2. To determine whether $v$ is kept in memory at a given time $t$, we use a function $L_S(v, t) \mapsto \{0, 1\}$. This function outputs 1 if $v$ is kept in memory at time $t$, and 0 otherwise.

$$L_S(v,t) := \begin{cases} 1 & v \text{ is a model input} \\ 1 & v \text{ is a model output} \land \forall t' > t, v \notin \text{output}(n_{t'}) \\ 1 & \exists t'' \geq t, v \in \text{input}(n_{t''}) \land \forall t' \in [t, t''), v \notin \text{output}(n_{t'}) \\ 0 & \text{otherwise} \end{cases}$$

Let $M(S)$ be the minimum memory size required to compute a valid sequence $S$. Then, we have $M(S) = \max_{1 \leq t \leq T} \sum_{v \in V} L_S(v, t) \cdot s(v)$. Also, we define the cost of the sequence, $C(S)$, by $C(S) = \sum_{1 \leq t \leq T} c(n_t)$.

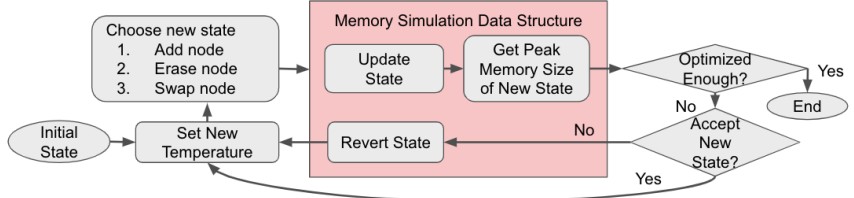

Figure 3: Overview of the simulated annealing.

**Recomputation**   Let $f$ be an objective function which maps sequences to a real number. The recomputation problem is defined as the construction of a valid sequence $(n_1, \ldots, n_T)$ that minimizes $f(S)$.

When minimizing memory usage, we set $f(S) := M(S)$. The objective function can depend not only on $M(S)$ but also on computational costs. In our experiments, we utilized Equation 1:

$$f(S) := \max(\text{budget}, M(S)) \times C(S) \tag{1}$$

This function minimizes the total computational cost when the memory budget is met. Otherwise, it tries to minimize both the peak memory and the cost.

### 2.1.1   Add-max segment tree

The range-add/range-max segment tree [5] is a data structure that can evaluate add and sum queries for intervals lazily. The add-max segment tree, utilized in this paper, holds an array $A[T]$ of length $T$, and can carry out the following operations in $O(\log T)$ time:

- Add $x$ to $A[i]$ for $i \in [l, r)$
- Obtain the maximum value of $A[i]$ for $i \in [l, r)$

### 2.2   Fast Simulated Annealing Algorithm

The proposed algorithm is based on simulated annealing (SA) with a straightforward outline. We make small modifications to the sequence and accept them if they decrease the objective function $f(S)$. Please see Figure 3 for an overview of the algorithm. Two critical mutations must be performed in sublinear time for the algorithm to be efficient: (1) Slightly modifying $S$ to create $S'$. (2) Computing $f(S')$.

To generate $S'$ from $S$, we either add or remove a node. We represent the sequence as a fixed-sized vector of nodes, where each position in the sequence corresponds to a specific time step. This representation makes it easy to manage the lifetime of each value. The vector contains a special node, nop, which has zero computational cost and no inputs or outputs associated with it. When adding or removing a node, its value is exchanged with that of nop.

We start with a valid sequence $S = (n_1, ..., n_T)$ and for all $1 \leq i \leq T$, $n_i \in \{\mathcal{N} \cup \text{nop}\}$. To enable the insertion of a long computation between nodes, most elements of $S$ should be initialized with nop. In practice, the initial $S$ is constructed from the sequence representing the initial execution order by inserting nop nodes between each element.

In each iteration, we perform one of the following three mutations:

1. (Add computation) Select a random node $n \in \mathcal{N}$ and a random time $t$ where $n_t = \text{nop}$. We attempt to update $n$, i.e., $n_t \leftarrow n$.

2. (Remove computation) Select a random time $t$ such that $n_t \neq \text{nop}$ and attempt to update $n_t \leftarrow \text{nop}$.

3. (Rotating sequence) Select two random times $t_1$ and $t_2$ such that $n_{t_1} \neq \text{nop}$ and $n_{t_2} = \text{nop}$. Try swapping $n_{t_1}$ and $n_{t_2}$. This modification can be implemented with the combination of the first two modifications.

To ensure that the mutated sequence $S'$ has a valid dependency, we validate the following conditions and update the sequence only if they are satisfied. Note that we can efficiently check these conditions by maintaining a set of produced and used times for each value $v$.

1. (Add computation) For each input $v$ of $n_t$, $v$ must be produced before time $t$.
2. (Remove computation) For each output $v$ of $n_t$, either $v$ must already be produced before time $t$, or there must not be any user of $v$ before the next production of $n_t$.

An important point to note is that the mutations 1 and 2 are inverses of each other, meaning that to undo a change, one simply needs to reapply the inverse mutation.

### 2.2.1 Updating peak memory and objective

In the following discussion, we consider the update of $f(S)$ by mutation 1 and 2. Since the objective function $f(S)$ often depends on both memory usage $M(S)$ and the overall cost $C(S) = \sum_t c(n_t)$, we assume this is still the case here. We can easily update $C(S)$ by adding or subtracting the corresponding cost of a newly added or removed node from $S$. Updating $M(S)$ is more challenging; to calculate the maximum peak memory consumption when a node is added or removed from $S$, we have to calculate the aggregated memory consumption by the values with overlapping lifetimes.

The lifetime of inputs and outputs change when adding or removing node $n$ (see Figure 5 in Appendix for illustration). However, the insertion and deletion of a node only slightly modify the life intervals for each node's input and output values. Consider a value $v$ with life intervals $L(v) := \{(t_i, t_j)\}$ determined by $L_S(v, t)$. Suppose node $n$ is inserted at time $t$, and $v$ is an input of $n$. We update the life intervals of $v$ as follows: If there is $(t_i, t_j) \in L(v)$ such that $t_i \leq t \leq t_j$, do nothing. Otherwise, take a $(t_i, t_j) \in L(v)$ such that $t_j \leq t$ is maximum (such $t_j$ exists if the sequence after insertion is valid). Update $L(v) \leftarrow L(v) \backslash (t_i, t_j) \cup (t_i, t)$. Similar rules apply for updating the life intervals of the outputs of the inserted node and the inputs and outputs of the node to remove. Specifically, on inserting or deleting a node, the update of life intervals for an input or output value applies to one of the following four cases:

1. No update.
2. Extend or shrink a range $(t_L, t_R)$ to $(t_L, t_{R'})$ or $(t_{L'}, t_R)$.
3. Split a range $(t_L, t_R)$ into $(t_L, t_{R'})$ and $(t_{L'}, t_R)$.
4. Merge two ranges $(t_L, t_{R'})$ and $(t_{L'}, t_R)$ into $(t_L, t_R)$.

Because the update of live intervals involves adding or subtracting the value size $s(v)$ from a certain range for each input or output value $v$ (as depicted in Figure 5 in Appendix), we can efficiently maintain memory usage and calculate the peak memory using a segment tree. This data structure allows for range-max and range-sum queries to be performed in $O(\log T)$ time, as introduced in Section 2.1.1. The segment tree maintains memory usage for time $t$ in an internal array $A[t]$, and peak memory can be determined efficiently by taking the maximum value in the range $[0, T)$ in $O(\log T)$ time. To be specific, we add $s(v)$ for the extended range and subtract $s(v)$ for the removed range when updating the live intervals. The segment tree significantly contributes to the simulated annealing as it can reduce the time for the differential update of a lifetime interval from the naive $O(T)$ time to $O(\log T)$ time.

### 2.2.2 Improving memory reduction by grouping

To determine an optimal recomputation sequence, it may be necessary to replicate specific node patterns to rematerialize a larger value. However, achieving this using random insertion and deletion of nodes is challenging. To address this issue, we introduce the concept of grouped nodes, which concatenates multiple nodes in a series to enable the recomputation of a series of nodes. A grouping node, represented by $g$, is formed by concatenating two nodes, $n_1$ and $n_2$, and has the following properties:

- $c(g) = c(n_1) + c(n_2)$
- $\text{input}(g) = \text{input}(n_1) \cup (\text{input}(n_2) \backslash \text{output}(n_1))$
- $\text{output}(g) = (\text{output}(n_1) \backslash \text{input}(n_2)) \cup \text{output}(n_2)$

Grouped nodes can be especially useful for sequenced patterns where $n_1$ must be computed immediately before $n_2$, as grouping these nodes together can make the sequence more likely to converge during simulated annealing. Additionally, we observed that after conducting annealing with grouped nodes, further improvement can be achieved by decomposing the grouped node and performing another round of annealing with lower temperatures. In Appendix C.1, we provide further discussion on the benefits of using node grouping for optimization.

### 2.3  Other Considerations

Conceptually, recomputation enables to optimize the time-space tradeoff without changing the graph semantics, i.e., the output values of the neural network. However, in real models, certain operators with internal states or side effects may produce different output values if recomputed or if their computation order is changed. Our algorithm can handle these cases by simply prohibiting the node addition or removal for these nodes.

An additional way to reduce memory usage that can be easily incorporated into our recomputation algorithm is offloading, which involves moving tensors from GPU memory to host memory when they are not immediately needed, and then moving them back before performing the dependent operation. Our algorithm can be extended to support offloading, as outlined in Appendix B.3. Supporting offloading together with recomputation offers the potential for even greater reductions in memory usage, particularly for larger computational graphs.

## 3  Experiments

In this section, we present the results of our experiments with the recomputation algorithm. Our algorithm was integrated into the PyTorch framework, and we used it to optimize the internal computational graph of various popular models, including vision and text architectures.

### 3.1  Configuration

**Model and input data**    Our experiments involved the latest vision models and vision transformers obtained from timm (PyTorch Image Models), as well as text models (including language models) from Hugging Face transformers. To obtain the full computation graph with backpropagation, the vision models were set up for image classification, and model variants with sequence classification heads for the text models. The computational graphs were obtained by PyTorch's symbolic tracing. The value sizes are estimated from shapes and data types. Since node cost estimations were not available in symbolic tracing, all nodes were estimated to have unit costs. For memory budgets, we used the 50% and 25% values of the simulated initial peak memory. For additional details on the environment, PyTorch integration, hyperparameters of the models (e.g., batch sizes, sequence length in transformers), and the effect of using simulated costs instead of actual ones, please refer to Appendix D.

**Objective function and hyperparamters of SA**    As outlined in Section 2, we utilized an objective function of the form $f(S) = \max(\text{budget}, M(S)) \times C(S)$ for our SA algorithm unless mentioned otherwise[2]. This objective function seeks to minimize $C(S)$ once the memory budget is met, and until that point, it minimizes both memory and cost. We included cost in the memory minimization objectives to address the risk of large numbers of nodes, which can result in slower convergence. This objective function is continuous and independent of the cost or size units, and it works effectively for a broad range of practical models.

To ensure efficient convergence, we utilized the first SA on grouped nodes for at most 20 million iterations, or until the memory budget was met, whichever occurred first. The second SA ran for a fixed 2 million iterations for the purpose of cost reduction as detailed in section 2.2.2.

**Checkmate**    To the best of our knowledge, the most powerful recomputation planner currently available is Checkmate [19]. We re-implemented Checkmate for PyTorch. However, Checkmate

---

[2] In certain scenarios, this objective function may exceed the budget constraint. For experiments of Table 1 and Table 2, where memory budgets must be satisfied, we employed $f(S) = L(\text{Elu}(L^{-1}(M(S)))) \times C(S)^{0.6}$, where $L$ is a linear function with $L(0) = \text{budget}$ and $L(-100) = 0$, determining the sharpness of Elu.

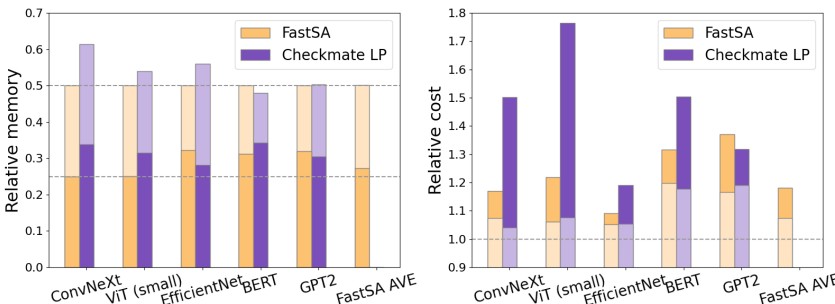

Figure 4: Comparison of simulated memory decrease and cost increase by recomputation. For each model, the memory budget is set to 50% and 25% of the simulated initial peak memory (darker bars represent the 25% budget and light bars represent 50% budget). FastSA AVE represents the geometric mean for all 23 models by FastSA. Checkmate LP's average bar is not shown since failed to solve 11 out of 23 model instances.

could not find any feasible solution within 6 hours for all models with more than 500 nodes due to the size of the integer linear programming (MILP) to be solved. Therefore, for the comparison with large models, we resort to Checkmate LP, where the MILP was relaxed to linear programming (LP). For clarification, the original version is referred as Checkmate MILP. Gurobi [15] is used as the internal solver for Checkmate MILP together with PDLP solver, provided by OR-Tools [28] for Checkmate LP[3].

**Moccassin**   Another alternative to perform recomputation is Moccasin [3]. Similarly to Checkmate, Moccasin uses constant programming (CP) to solve the recomputation problem. It introduces a new hyperparameter that acts as an upper-bound on how many times a value can be recomputed. Thanks to this limitation, the number of integer variables involved in the CP set is linear to the number of nodes in the graph, allowing a faster execution time than Checkmate. However, the possible solutions that Moccasin can converge to are heavily limited by this hyperparameter value and the achievable memory reductions can be sub-optimal.

## 3.2   Memory reduction on large models

Figure 4 provides a comparison between FastSA and Checkmate LP in terms of simulated memory reduction and cost increase. We selected the most representative models for this figure, in addition to the geometric mean of all 23 models tested in Appendix D.7. Checkmate LP failed to find solutions for 11 models due to time limits or out-of-memory errors of 100 GiB, so the geometric mean is not reported. Moreover, memory usage was not fully reduced to the budget due to LP relaxation and randomized rounding, which resulted in larger overhead for some models, such as ViT (small); we discuss this behavior in Appendix D.6.

For the 50% memory budgets, FastSA algorithm was successful in meeting the target budget for all models, with an average increase in model execution time of 7%. However, for the 25% budges, FastSA could not reduce memory to the budget for some models due to either limitations on the model itself or convergence to a suboptimal solution due to SA heuristic limitations. Despite this, the algorithm reduced memory usage by an average of 73% with an average increase in overhead of 18%. Overall, these results demonstrate the superior performance of FastSA compared to Checkmate LP in reducing memory usage, particularly for larger neural networks.

A comparison with Moccasin was also conducted by applying FastSA to the publicly available data in [3]. Table1 presents the results from the original Moccasin study, extended to include outcomes for FastSA. The experiments pertain to two graph variants, namely random-layered (RL) and graphs mentioned in [19] (CM). For the RL graphs, FastSA achieved the lowest recomputation overhead

---

[3]The PDLP solver is used for Checkmate LP due to the licensing costs associated to Gurobi. Checkmate LP experiments were executed in parallel using a cluster environment. We verified that for the experiments in Figure 4, the PDLP solver could solve all the instances solved by Gurobi, making it a reasonable open-source alternative.

| Graph | $(n, m)$ | B | CHECKMATE MILP | | | MOCCASIN | | | FASTSA | | |
|---|---|---|---|---|---|---|---|---|---|---|---|
| | | | CI | Mem | Time | CI | Mem | Time | CI | Mem | Time |
| RL 1 | 100 | 90 | 0.8 | 89.2 | 18.5 | 0.8 | 88.1 | 9.3 | **0.0** | 79.4 | 11.7 |
| | 236 | 80 | 2.3 | 79.5 | 22.7 | 2.3 | 79.5 | 9.5 | **0.3** | 79.4 | 11.5 |
| | | 70 | no experiment | | | no experiment | | | **2.2** | 77.3 | 12.0 |
| RL 2 | 250 | 90 | 0.9 | 90.0 | 685.1 | 0.9 | 89.8 | 55.0 | **0.0** | 77.3 | 15.1 |
| | 944 | 80 | time limit exceeded | | | 4.9 | 80.0 | 639.5 | **0.0** | 72.2 | 15.0 |
| | | 70 | no experiment | | | no experiment | | | **2.6** | 68.9 | 14.5 |
| RL 3 | 500 | 90 | time limit exceeded | | | 0.7 | 90.0 | 1803.3 | **0.03** | 87.4 | 21.0 |
| | 2461 | 80 | time limit exceeded | | | 3.4 | 80.0 | 1804.8 | **2.3** | 78.6 | 20.9 |
| | | 70 | no experiment | | | no experiment | | | **4.8** | (73.5) | 21.2 |
| RL 4 | 1000 | 90 | time limit exceeded | | | 0.7 | 90.0 | 3612.9 | **0.4** | 87.5 | 36.0 |
| | 5857 | 80 | time limit exceeded | | | 3.4 | 80.0 | 3611.8 | **2.5** | 78.4 | 36.3 |
| | | 70 | no experiment | | | no experiment | | | **7.4** | 70.0 | 36.5 |
| CM 1 | 73 | 90 | 0.0 | 88.4 | 6.3 | 0.0 | 88.4 | 3.1 | **0.1** | 75.3 | 9.4 |
| | 149 | 80 | 0.1 | 76.9 | 5.6 | 0.1 | 78.9 | 3.1 | **0.1** | 75.1 | 9.7 |
| | | 70 | no experiment | | | no experiment | | | **3.0** | 62.3 | 9.7 |
| CM 2 | 353 | 90 | 0.1 | 89.0 | 434.1 | 0.2 | 89.9 | 65.2 | **0.2** | 86.2 | 10.9 |
| | 751 | 80 | 0.3 | 79.7 | 485.3 | 0.3 | 80.0 | 69.3 | **0.4** | 76.6 | 10.9 |
| | | 70 | no experiment | | | no experiment | | | **0.8** | 66.8 | 11.0 |

Table 1: **Comparison with Moccasin.** The table includes results of Checkmate MILP and Moccasin from [3], extended with FastSA data. The columns B and CI represent memory budget and cost increase percentage, respectively. Alongside original results for 90% and 80% budgets, a 70% budget row only demonstrating FastSA results is also inserted. For all random-layered (RL) cases, FastSA exhibited the smallest CI, as it optimizes topological ordering, reducing memory without materialization.

in all instances. Remarkably, for RL1 and RL2 cases, FastSA managed to decrease memory usage without adding new recomputation nodes, through the optimization of the execution's topological ordering.

## 3.3 Solution optimality

| Model | $(n, m)$ | B | CHECKMATE MILP | | | FASTSA | | |
|---|---|---|---|---|---|---|---|---|
| | | | CI | Mem | Time | CI | Mem | Time |
| VGG11 | (69, 119) | 90 | 1.4 | 87.6 | 5.1 | 2.9 | 87.5 | 2.4 |
| | | 80 | 2.9 | 79.8 | 1.5 | 2.9 | 79.8 | 2.4 |
| | | 70 | infeasible | | | 2.9 | (79.8) | 2.4 |
| ResNet18 | (171, 437) | 90 | 0.6 | 85.7 | 47.6 | 0.6 | 85.7 | 2.1 |
| | | 80 | 1.8 | 78.6 | 45.3 | 2.9 | 75.0 | 2.1 |
| | | 70 | 2.3 | 67.9 | 367.3 | 4.7 | 66.1 | 2.1 |
| | | 60 | 4.1 | 57.1 | 1846.0 | 5.3 | 55.4 | 2.2 |
| | | 50 | (4.7) | (50.0) | > 3600 | 6.4 | (51.8) | 2.3 |

Table 2: **Comparison between FastSA and Checkmate MILP on small models.** For the resnet18 case with 50% budget, the best feasible solution was used because the MILP solver could not find the optimal solution within the time limit.

We conducted a study on how closely FastSA solutions match with the optimal solution by comparing its results with those of Checkmate MILP on small models. The results, compiled in Table 2, show that for the model VGG11 under the strictest budget constraint (80%), FastSA located the same plan as Checkmate MILP for recomputation. For ResNet18, except for a 50% budget, FastSA managed to cut down memory use under the budget in all cases, but with up to 2.4% more overhead from recomputation compared to Checkmate MILP. Even though FastSA fell short of finding the optimal solutions for these cases, it was up to 1000x faster than Checkmate MILP, particularly

under strict memory budget scenarios. In certain situations where the computational graph has high topological freedom, Checkmate might present suboptimal recomputation plans (discussed in Appendix A.1). This can be found in the results for RL graphs in Table 1, where FastSA could find better solutions than Checkmate MILP or Moccasin.

## 4 Related Work

### 4.1 Model Parallelism

One common approach to scaling a single neural network that is limited by memory is to partition it into multiple devices using either Model Parallelism or Pipeline Parallelism. Model Parallelism [34] involves horizontally splitting the neural network by distributing the parameters of each layer across multiple GPU devices, while Pipeline Parallelism [26] proposes to vertically split the network by assigning each device several contiguous layers of the model. While these approaches enable deep neural networks to be trained at scale, their performance can be limited by the communication overhead required for each iteration of the model. Rajbhandari et al. [30] found that these approaches can achieve only 5% of the performance peak of a V100 device, highlighting the limitations of these techniques. These limitations have spurred further research into improving neural network scaling, which is discussed in more detail below.

### 4.2 Recomputation

Recomputation is a technique that was first introduced in classical compiler research to minimize the number of required registers, and later on was adapted for use in deep neural networks (DNNs) by Chen et al. [7] as a means of reducing memory consumption during training of sequential models. However, this method is limited to sequential graphs and disregards node costs. Kusumoto et al. [21] proposed dynamic programming algorithms for more general computational graphs. Also, Kumar et al. [20] leverages tree decomposition to handle more general graph structures. However, these methods still requires large computational overhead, making them impractical for larger networks.

Jain et al. [19] formulated recomputation as a mixed integer linear programming (MILP) problem and proposed Checkmate, a solver to find an optimal recomputation plan. Checkmate tries to minimize the total computational cost under memory budget and dependency constraints. Although Checkmate has shown to significantly outperform existing methods in terms of solution quality, it requires substantial computational resources to solve the MILP. The number of decision variables in the MILP scales quadratically to the graph size and time scales exponentially to it. To address the limitation of Checkmate, Bartan et al. [3] proposed Moccasin, which formulates recomputation using constraint programming (CP). The number of integer variables is reduced from quadratic to linear in Moccasin, by setting a hyperparamter for the maximum times a value can be recomputed, thus expediting execution. However, this formulation also narrows the search space and may impact overall quality of the recomputation plans when compared to Checkmate or FastSA. Also, in a parallel development, Rockmate [44] was proposed for models with repeated layers. It decomposes the problem of recomputation into intra-layer and inter-layer recomputation. It applies Checkmate to a single layer, which forms a smaller graph than the entire model, and then finds the recomputation plan across the layers by Rotor [17], a dynamic programming based recomputation algorithm that works for sequential models.

In contrast, our proposed method converges to solutions that are comparable or even better than Checkmate and Moccasin with a single CPU core, taking less than four seconds on average. This significant reduction in computational time is achieved by leveraging efficient heuristics and optimization techniques. Our results demonstrate that our approach has great potential in the context of real-world applications, especially for cases where large computational resources are not available.

### 4.3 Other techniques for memory reduction

There are several other techniques that have been proposed to reduce device memory consumption in addition to recomputation, including offloading, which involves transferring some of the model parameters to a system's CPU or an external memory device when they are not immediately required. Beaumont et al. [4] discuss the combination of offloading and recomputation as a means of further reducing memory utilization. Our proposed method can easily be extended to support

offloading, as detailed in appendix B.3. Another approach that leverages offloading is the ZeRO technique proposed by Rajbhandari et al. [30], which partitions the model parameters, optimizer states, and activations among several devices to increase parallelism and reduce the memory used by each device. This approach enables the training of exceptionally large models with hundreds of billions of parameters, making it a powerful tool for advanced natural language processing and computer vision applications. Other techniques focus on reducing the size of the parameters and intermediate results, such as quantization [23], which reduces the floating point precision of computations and weight storage up to 2-bits in extreme cases. Sparsification [25] exploits the sparsity patterns that arise during computation to reduce the total needed memory while keeping the same precision. There are also more exotic approaches, such as the reversible residual network [14], which is a memory-efficient architecture that can perform backward computation without saving the activations. However, its applicability is limited to residual networks only. It is worth noting that these techniques are largely orthogonal to our proposed method and can be combined to further improve memory savings in neural network optimization.

## 5  Conclusion

In this paper, we present a novel method for recomputation that offers several key advantages over existing approaches. Our method is applicable to general graphs and can support any objective function that depends on peak memory usage and the total cost of computation. Moreover, it can find near-optimal recomputation plans within a remarkably short computational time of 3 to 30 seconds, even for large computational graphs.

Another major advantage of our method is its efficiency, as it is single-threaded and uses memory resources in a highly efficient manner. This makes it ideal for integration into neural network compilers, where it can further streamline the optimization process. Additionally, our algorithm is highly flexible and can handle a wide range of problem settings, including recomputation with offloading and consideration of node-intermediate memory usage.

We explored the tradeoff between computation time and memory usage and evaluated the effectiveness of our algorithm in terms of reducing peak memory usage. Our experiments demonstrate that our approach can achieve significant reductions in peak memory usage, with reduced impact on computation time, for a wide range of neural network architectures. Overall, our experiments validate the effectiveness and practicality of our recomputation algorithm, highlighting its potential for widespread application in many areas of deep learning research and development.

**Limitations**  In conclusion, while our algorithm offers a powerful solution for complex neural network optimization problems, it has certain limitations that must be considered. As demonstrated in Figure 4, our approach was able to reduce memory usage more significantly than Checkmate LP in most cases, but the result was not ideal for some models due to suboptimal node grouping.

When planning recomputation in distributed training, the computational graph may need to have extra requirements such as (1) the optimized graph must be able to split equally for each worker in Pipeline Parallel or (2) the communication operators must be overlapped with arithmetic computations. Currently, it is difficult to handle these constraints with the standard FastSA.

Additionally, the peak memory estimation may not be always precise due to memory aliasing, caused by operators that change the memory views or do inplace updates. This is one of the reason why there is a gap between the simulated and actual GPU memory usage as shown in Appendix D.8.

Overall, our proposed approach offers a fast and efficient method for neural network optimization, providing significant improvements over existing techniques. However, future research may be required to support a wider variety of computation.

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

# A Checkmate

As previously introduced in the main section, Checkmate is a recomputation planner that operates using mixed integer linear programming. This appendix provides additional details on the internal workings of Checkmate.

**Algorithm** Let $(n_1, \ldots, n_T)$, where $n \in \mathcal{N}$, be the original computation order. To define the $i$-th stage of computation ($1 \leq i \leq T$), Checkmate uses a $T \times T$ matrix $\{R_{i,j}\}$, where $R_{i,j} \in \{0, 1\}$. When given a matrix $R$, the $i$-th stage of computation is a subsequence of the original computation order, defined by $(n_j \mid R_{i,j} = 1)$. The recomputation plan is then determined as the concatenation of $T$ stages. The maximum length of the recomputation sequence is $O(T^2)$ as a result of this formulation.

The objective of Checkmate is to minimize $\sum_{i=1}^{T} \sum_{j=1}^{T} R_{i,j} c(n_j)$, where $c(n_j)$ represents the cost of recomputing node $n_j$. Additionally, Checkmate uses binary variables to determine whether to preserve the outputs of $n_j$ from stage $i$ to $i+1$ and whether to free the outputs of $n_j$ in stage $i$ after evaluating $n_k$. The number of binary variables is $O(|\mathcal{N}|^2 + |\mathcal{N}|M)$, where $M$ is the number of arcs in the DAG. Checkmate also introduces variables to represent the memory usage during the computation of node $n_j$ at each stage. Lastly, the memory budget constraint and the dependency constraints among the variables are added to formulate the recomputation problem as a mixed-integer linear program (MILP).

## A.1 Limitations of Checkmate

Despite the ability of Checkmate to consider node and value weights for any computation graph, it has several limitations.

**Fixed topological order** One such limitation is the requirement for a fixed topological order. When a computational graph has multiple possible computation orders (i.e., the topological order is not unique), the memory usage can differ significantly based on the computation order selected. Identifying the topological order that minimizes memory usage is a difficult problem [33], and Checkmate may not be able to reduce memory usage significantly if the initial computation order is not memory efficient. As feasible recomputation sequences can become exponentially long for some graphs and budgets [27], it is challenging to increase the number of stages to enable Checkmate to accommodate any feasible computation order.

**LP-related limitations** There are several additional limitations to Checkmate when the MILP is relaxed to linear programming (LP). Checkmate may fail to provide optimal solutions due to MILP/LP gaps, randomized rounding, and solver tolerances. In most cases, the LP relaxation method results in better objective values than the original MILP by allowing non-binary assignments for binary variables. However, Checkmate then applies randomized rounding on the binary variables that determine whether the outputs of $n_j$ at stage $i$ is kept for stage $i+1$, which can increase the objective value. The original paper suggests setting $0.9 \times \text{budget}$ for the LP relaxed problem, considering the memory usage increase after randomized rounding. While often there is no significant degradation in memory consumption, execution time may unexpectedly increase due to this process. We have observed that since the number of elements in $\{R_{i,j}\}$ is $O(|\mathcal{N}|^2)$, it is less likely to have short recomputation sequences via randomized rounding when $|\mathcal{N}|$ is large. The degradation in the recomputation cost is more likely to occur when the LP solver's tolerances are not small enough. However, determining the solver tolerances becomes a trade-off between convergence condition and optimization time.

# B Application to extended problem settings

## B.1 Computational graphs with node-intermediate memory usage

In this section, we explain some of the extended problem settings where our algorithm can be applied.

The original description in Section 2 considers only the memory usage of input and output values. This setting is reasonable for two reasons: first, deep learning frameworks and neural network compilers have shape propagation, making it easy to infer the sizes of intermediate tensors. Second, in most cases, peak memory usage can be simulated with high precision even if node-intermediate memory usages are ignored.

However, precise peak memory estimation is necessary if the recomputation sequence must be highly optimized. For instance, if there is only a scarce amount of tensors alive when the memory usage reaches its peak, we must consider node-intermediate memory usage; otherwise, the estimation error will be significant. Additionally, certain operators, such as matrix multiplication, temporal values may consume a large amount of memory space. Considering node-intermediate memory usage is exceptionally important when dealing with grouped nodes. A grouped node may have substantially more node-intermediate memory usage when it involves a lot of computation. In our implementation, we estimate the node-intermediate memory of each grouped node from the temporal values produced and used within it.

**Extensions in algorithm**   In the algorithm stated in Section 2.2, when a node $n$ is added or removed from time $t$, we need to update the lifetime intervals for its input values and output values. To consider node intermediate values, we need an additional update: add (or subtract for removal) $s'(n)$ at time $t$, where $s'(n)$ denotes the intermediate memory usage of $n$.

## B.2   Memory minimizing topological ordering

Our algorithm can also be used to find a memory-minimizing computation order without recomputation. As mentioned in Appendix A.1, the problem of finding a topological order of a DAG that minimizes peak memory is known to be NP-hard. We can use our algorithm to heuristically solve this problem by prohibiting the addition and removal of nodes and allowing only node rotation during mutations in the SA.

## B.3   Offloading

Offloading, along with recomputation, is a popular method for reducing GPU memory usage. Offloading involves moving data to secondary memory and back when it is needed. Our algorithm is capable of optimizing offloading and recomputation simultaneously (although it could be applied to offloading alone).

The extended problem setting is as follows:

Given a computational graph and objective function $f$, the task is to construct a valid sequence $S = (n_1, \ldots, n_T)$ and fetching sequence $(F_1, \ldots, F_T)$, where $F_i \subseteq \mathrm{input}(n_i)$, that minimize the objective function. Here, $F_i$ denotes which input values are fetched (instead of using previously produced ones) when computing $n_i$. If all $F_i$ are set to empty, the problem is equivalent to the original recomputation. Note that the objective function $f$ is now dependent on both $S$ and the fetching sequence. For example, we can define the computational cost as $\sum_{i=1}^{T}(c(n_i) + \sum_{v \in F_i} c'(v)) + \sum_{v \in \{v | \exists i. v \in F_i\}} c''(v)$, where $c'(v)$ and $c''(v)$ denote the cost of fetching and offloading, respectively. This objective function assumes that offloading and fetching are done sequentially during computation. If background offloading and fetching are allowed, another suitable objective function may be defined.

Note that we only consider the timings of fetching in this problem setting, as the optimal timings of offloading can be determined when they are given.

**Extensions in algorithm**   We initialize all $F_i$ to be empty and allow an additional mutation in each SA iteration.

4. (Flip fetch) Choose a random time $t$ such that $n_t \neq \mathrm{nop}$ and random value $v \in \mathrm{input}(n_t)$. If $v \in F_t$, update $F_t \leftarrow F_t \backslash \{v\}$. Otherwise, update $F_t \leftarrow F_t \cup \{v\}$.

Unlike existing three mutations, this mutation is always valid, i.e., does not break the dependency of the sequence. Also, the inverse of this mutation is the mutation itself. The update of the life intervals

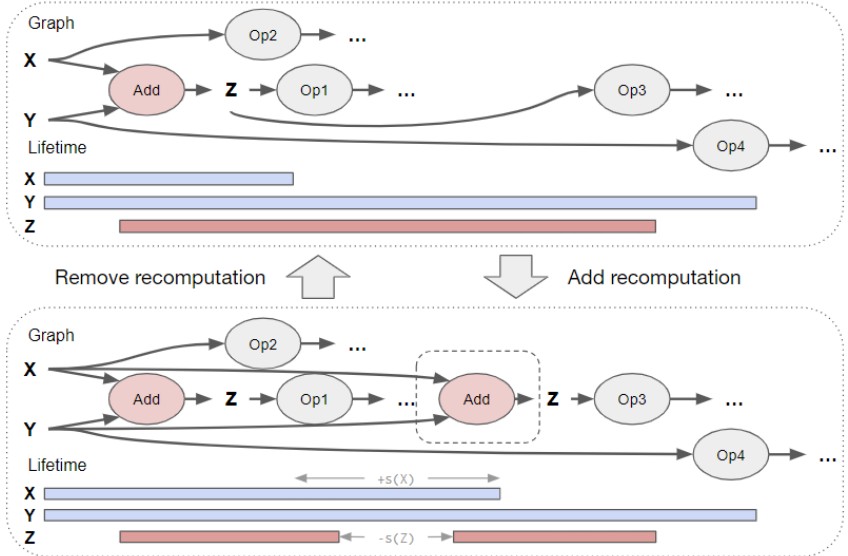

Figure 5: The update of lifetime by node addition or node removal. By adding a recomputation node, the life intervals of X and Z change. For either case, the update of memory usage can be easily managed by the range-add / range-max segment tree.

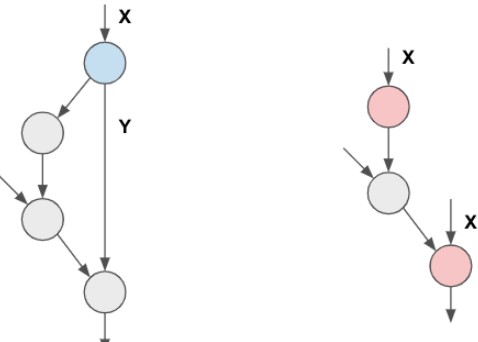

Figure 6: Example of node grouping. The blue node in the left graph is fused together with its successors. The red nodes in the right graph are the fused nodes taking $X$ as the input. Even without the SA, node grouping itself can optimize recomputation since the last value will be always recomputed from $X$.

of $v$ can be done in the same way as other mutations by considering the fetching and offloading as the addition or removal of a special node which outputs $v$ from empty inputs.

## C    Technique for fast and better convergence

This section presents various techniques to expedite the convergence of SA or obtain better solutions.

### C.1    Node grouping

As mentioned in Section 2.2, node grouping is a crucial pre-processing step for recomputing a sequence of nodes. Without node grouping, the SA has small chances of reaching an solution that effectively reduces the cost of the target function.

Figure 6 provides an example of node grouping. Even without the SA, node grouping can be used as a simple recomputation strategy to lower peak memory usage. In practice, our implementation uses the following algorithm. The do_fuse($n_i$) function in Algorithm 1 determines whether to fuse

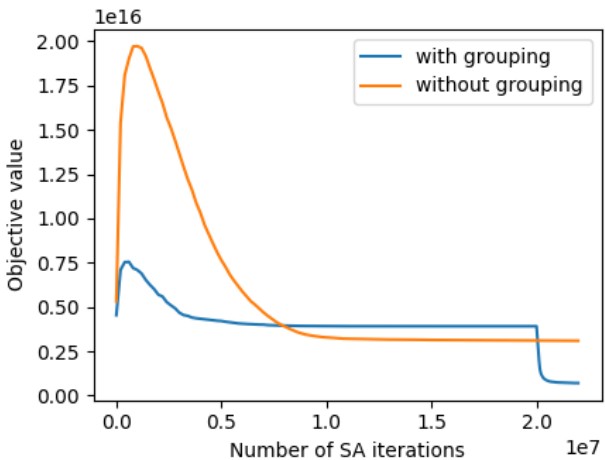

Figure 7: Memory reduction using node grouping for the LLaMA model. The blue line is the optimization with node grouping with the last 2M iterations being for decomposed nodes. The orange line is the SA without node grouping. The objective function is defined by equation (1) and the memory budget 6.7e+10, which is 10% of the original memory consumption.

a node $n_i$ with its succeeding operators. We perform the fusion if the outputs of $n_i$ can be computed from inputs with a smaller memory footprint, i.e., $\sum_{v \in \text{input}(n_i)} s(v) \leq \sum_{v \in \text{output}(n_i)} s(v)$.

---
**Algorithm 1** Node grouping
---
**Require:** Valid computational sequence $S = (n_1, \ldots, n_T)$.
**Ensure:** $S$ is a valid sequence with grouped nodes.
 1: **for** $i = 1, \ldots, T$ **do**
 2:     **if** `do_fuse`$(n_i)$ **then**
 3:         **for** $j = i + 1, \ldots, T$ **do**
 4:             **if** $\text{output}(n_i) \subseteq \text{input}(n_j)$ **then**
 5:                 $n_j \leftarrow$ grouped node of $n_i$ and $n_j$.
 6:             **end if**
 7:         **end for**
 8:         $n_i \leftarrow \text{nop}$
 9:     **end if**
10: **end for**
---

The node grouping algorithm with the `do_fusion` function mentioned above has several characteristics. Firstly, Algorithm 1 does not depend on the initial topological order of the computational graph and therefore works even if the original sequence is not memory-efficient enough. Secondly, the above node grouping algorithm can optimally reduce the memory usage in specific simple cases, including the training graphs of sequential models, which were discussed in early studies of recomputation techniques by Chen et al. [7]. In fact, the above node grouping algorithm can reduce the peak memory of such graphs to a constant size providing that the all the values have the same size.

Figure 7 demonstrates the importance of node grouping in recomputation for LLaMA. When node grouping was not used, the SA converged to a final objective value 4.4 times larger than the result obtained by the SA with node grouping. On the other hand, using only node grouping (without any SA), we were able to reduce memory usage for LLaMA to 7.8e+10, which is close to the 6.7e+10 budget, and lower than the optimized memory usage of 3.5e+11 achieved without node grouping.

### C.2 LogAddExp segment tree

In our optimization approach, we assume that the objective function is a function of peak memory $M(S)$ and computational cost $C(S)$. For each mutation in the SA, the cost may be updated, but

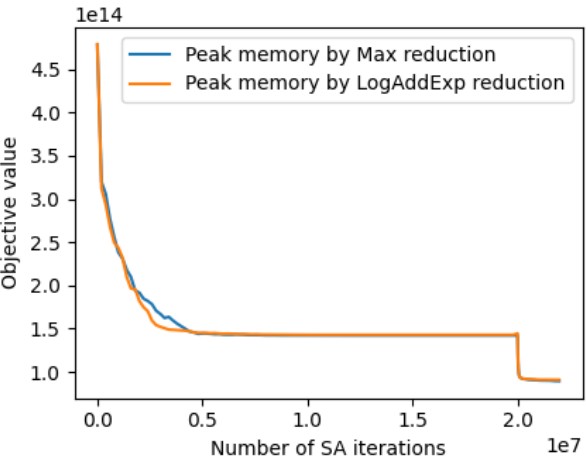

Figure 8: SA convergence for ViT model. We ran the first SA on grouped nodes for 20M iterations and the second SA on decomposed nodes for 2M iterations.

the peak memory $M(S)$ changes only when the mutation affects the peak timings. Suppose there is a computational graph with two memory peaks. In this case, $M(S)$ is determined by the higher of the two peaks, and mutations that reduce the memory usage around the second peak do not change $M(S)$. In general, multiple local memory peaks make it challenging to effectively reduce the objective value.

We can overcome this issue by employing range-add/range-logaddexp segment trees instead of range-add/range-max segment trees, introduced in Section 2. The range-add/range-max segment tree recursively calculates $\max(a, b)$ for a given interval and determines the maximum value within a range $[l, r)$. On the other hand, the range-add/range-logaddexp segment tree uses the recursive calculation of $\mathrm{logaddexp}(a, b) := \log(\exp(a) + \exp(b))$ instead of taking the maximum. The logaddexp segment tree takes into account multiple memory peaks, enabling faster convergence of the SA.

**Advantages and disadvantages**  While the logaddexp-based segment tree is expected to have better convergence when memory consumption is multimodal, it is essential to be wary of errors in memory usage simulation and numerical overflows. To avoid overflowing floating point operations, it is crucial to select an appropriate base for $\log$ and $\exp$. In the implementation used for Figure 8, we re-scaled value sizes so that the maximum value is 10 and set the base to 2. However, this configuration may still cause overflow, particularly when $|\mathcal{N}|$ is large. There are important considerations to be made regarding the values held by the segment tree. In a range-add/range-max segment tree, the $t$-the element of the segment tree represents the memory consumption at time $t$. However, in a logaddexp-based segment tree, this approach cannot be directly used since it leads to significant errors in calculating the peak memory. Specifically, unlike the max operator, $\mathrm{logaddexp}(x, x) = \log 2 + x > x$, meaning that repeatedly calculating $\mathrm{logaddexp}(a, b)$ for values $a$ and $b$ close to zero will result in increasingly large values. In our implementation, we utilized the fact that the peak memory is only achieved for time $t$ where $n_t \neq \mathrm{nop}$. We initialized the elements of the segment tree with a large negative constant and added a large constant again when a node is added at that time, ensuring more precise estimation of the peak memory.

Figure 8 shows the difference of the convergence by the kind of segment tree used in the SA state. The segment tree using logaddexp had slightly better convergence for the ViT model.

### C.3  Implementation of segment tree

The role of the segment tree used in our algorithm is to efficiently calculate the peak memory, which can be obtained by the range-max query. Here, we introduce another implementation variant of the segment tree.

**Dynamic segment tree**    The dynamic segment tree is a data structure that can efficiently handle the following operations: (1) Insert a value at index $i$. (2) Remove the value at index $i$. (3) Add value $x$ for range $[l, r)$. (4) Calculate the maximum value for range $[l, r)$. This kind of data structure can be created by extending binary search trees (for details, refer to textbooks such as [9]). By using this segment tree, we do not need $nop$, introduced in Section 2.2, and can optimize a computation sequence consisting of only $\mathcal{N}$. However, the dynamic segment tree often involves heavy implementation, so the actual optimization time could increase. On the other hand, it is more efficient in terms of memory as it requires $O(T)$ space, where $T$ is the maximum length of the computational sequence during optimization.

# D    Experiments

In this section, additional information for the experiments done throughout the paper is provided together with details on how to obtain computational graphs and the hyperparameters used to configure both, FastSA and Checkmate.

## D.1    PyTorch Computational Graph

**Optimizing PyTorch computational graph**    We extracted the computational graph using PyTorch's symbolic tracing mechanism, which is available for PyTorch 2.0 or later. For our experiments, we used PyTorch 2.1.0.dev20230404 as it had the necessary features for tracing unavailable in the latest stable release (v2.0.0).

Our recomputation algorithm was integrated by extending `aot_module`. This module acts as a wrapper of a model (`torch.nn.Module`) and optimizes the model's computational graph before the actual computation is done. Using symbolic tracing without real data, `aot_module` generates the computational graph for the model's forward and backward passes. It then combines them into a joint graph before optimizing and partitioning it for a new model with optimized forward and backward computations. Users can pass a custom partition function (an optional argument) to `aot_module` to take over the optimization process.

We implemented a custom partition function that follows these steps: it receives the joint graph and applies the recomputation algorithm to create a new computation graph that produces the same outputs from the same inputs. It returns the entire new computation graph as the forward module and configures the backward module to return only the pre-computed outputs assuming that all the tangents (i.e., gradient inputs) are one. The full graph is executed instead of separating it into forward and backward passes as it is more memory-efficient. When the graph is partitioned, some intermediate values for the backward pass need to be saved, and they remain in memory until the backward pass is complete.

**Preprocess for PyTorch intermediate graph representation**    We perform Dead Code Elimination (DCE) before applying the recomputation algorithms. Although our algorithm automatically does DCE during the SA by removing nodes, we perform explicit DCE in advance due to two reasons: dead code can affect node grouping performance and Checkmate does not support DCE.

Next, we convert the PyTorch intermediate graph representation (`torch.fx.Graph`) to an internal graph format with explicit nodes $\mathcal{N}$ and values $\mathcal{V}$. In `torch.fx.Graph`, the computational graph is represented as a list of nodes, and each value corresponds to a single node. If an operator has multiple output values in `torch.fx.Graph`, it is represented as a single node that outputs a tuple, followed by get-item nodes that extract specific values from the tuple using an index. We can apply the recomputation algorithm to `torch.fx.Graph` intermediate representations, but it is easier to convert `torch.fx.Graph` to a more general graph format to handle operators with multiple outputs efficiently. After the recomputation is done, we convert our format back to `torch.fx.Graph`.

For Checkmate, we follow the original paper's implementation and perform optimization directly on a `torch.fx.Graph` styled computational graph.

## D.2 Checkmate settings

We re-implemented Checkmate [19] for the PyTorch intermediate representation. For Checkmate MILP, we used Gurobi as the internal MILP solver. For Checkmate LP, we employed PDLP solver, provided by OR-Tools [28] to solve large-scale LP problems in a multithreaded manner. PDLP was the fastest solver among the available options (CLP, GLOP, and SCIP) in OR-Tools.

The execution time and solution quality of Checkmate LP are highly dependent on the hyperparameters of the LP solver, as discussed in Appendix A.1. We adopted the default values of 1e-4 for primal tolerance and dual tolerance in Checkmate's open-source implementation. We randomly generated 100 different thresholds in addition to the default threshold 0.5 for randomized rounding and selected the best solution as the final output after testing these thresholds.

## D.3 Environment

The proposed algorithm (FastSA) and Checkmate LP were evaluated using a cluster system, with each instance configured with 8 CPU cores (Intel(R) Xeon(R) Platinum 8380 @ 2.30GHz) and 100 GiB of RAM with a NVIDIA A100 80GB GPU. Although the PDLP solver fully uses the allocated resources, FastSA only requires a single CPU. Due to Gurobi license constraints, Checkmate MILP was executed on another machine with 36 CPU cores (Intel(R) Xeon(R) Gold 6154 CPU @ 3.00 GHz) and 376 GiB of RAM. The implementation of both FastSA and Checkmate was written in C++ and compiled using GCC 10.4.0 with the O3 option. The compiled module was integrated with PyTorch 2.1.0.dev20230404 and the experiments were run using Python 3.9.12 with CUDA 11.8.

## D.4 Settings of our algorithm

**Initialization of SA state** We initialized the default computation sequence $S$ used in our SA as follows. First we fix the length of $S$ as $T = 2^{20}$ and initialize all of them by nop. Then, for each $n_i \in \mathcal{N}$ of the default computation sequence $(n_1, \ldots, n_k)$, we set $S[t] \leftarrow n_i$, where $t = \frac{i}{k+1}T$.

**First-stage SA** The SA algorithm in the first stage is applied to grouped nodes for efficient memory usage reduction. The number of SA iterations is set to 20 million in this stage, but if the memory budget is met, the algorithm stops even if the temperature is still high.

The initial temperature is set as $0.1\%$ of the initial objective value, and the final temperature is set as $0.01\%$ of the initial temperature. The temperature in the $i$-th iteration is calculated as $h_i \times \exp(\log(h_f/h_i) \times i/N)$, where $h_i$ and $h_f$ denote the initial and final temperature, respectively, and $N = 2 \times 10^7$ is the maximum number of iterations.

**Second-stage SA** The second stage of the SA algorithm is applied to decomposed nodes and targets to remove redundant computations. The number of iterations in this stage is 2 million. The initial temperature is set as the final temperature in the first-stage SA. The final temperature is set as $0.01\%$ of the initial temperature, as in the first stage. The calculation of the temperature for each iteration is also the same.

## D.5 Models

The models are obtained from `timm` (PyTorch Image Models) and `transformers` provided by Hugging Face. The versions of timm and transformers are 0.9.1 and 4.28.1, respectively. All the models are set to the training mode. Random tensors are used for the inputs. For vision models shown in Table 3, we set batchsize as 512. For GPU benchmarks, we reduced the batchsize to 256 to compare the performance to the baseline, without recomputation. We summarize the all the models used in our experiments with their configurations below.

For text models shown in Table 4, we used batchsize 128 and context length 512 to obtain the simulated memory and computational cost after applying recomputation. For the real GPU benchmarking, we reduced the values to 64 and 256, respectively.

For language models shown in Table 5, we used batchsize 8 and context length 2048, except for GPT2 with maximum context length 1024 to obtain the simulated values. For the real GPU benchmarking, we reduced the values to 4 and 1024 for all of them.

| name | full model name | resolution | $|\mathcal{N}|$ | $|\mathcal{V}|$ |
|---|---|---|---|---|
| ConvNeXt [24] | convnext_tiny | 224 | 932 | 1253 |
| ConvNeXt V2 [42] | convnextv2_large | 224 | 2840 | 3467 |
| EVA-02 [13, 37] | eva02_large_patch14_224 | 224 | 6205 | 6967 |
| ViT [36, 12] | vit_large_patch16_224 | 224 | 2817 | 3362 |
| ViT (small) | vit_small_patch16_224 | 224 | 1425 | 1706 |
| MobileNetV3 [18, 41] | mobilenetv3_large_100 | 224 | 484 | 1155 |
| EfficientNet [38, 41] | efficientnet_b0 | 224 | 996 | 1766 |
| DeiT III [39] | deit3_base_patch16_224 | 224 | 1545 | 1850 |
| XCiT [1] | xcit_tiny_12_p16_224 | 224 | 2818 | 3550 |
| BEiT [2, 12] | beit_base_patch16_224 | 224 | 1835 | 2187 |
| CoAtNet [10] | coatnet_2_rw_224 | 224 | 2204 | 3128 |
| VGG11 [35] | vgg11 | 224 | 69 | 119 |
| ResNet18 [16] | resnet18 | 224 | 171 | 437 |

Table 3: List of timm models (PyTorch Image Models). $|\mathcal{N}|$ and $|\mathcal{V}|$ denote the numbers of the nodes and values of the computational graph.

| name | full model name | $|\mathcal{N}|$ | $|\mathcal{V}|$ |
|---|---|---|---|
| ALBERT [22] | albert-base-v2 | 2285 | 2419 |
| BERT [11] | bert-base-uncased | 2078 | 2386 |
| DistilBERT [31] | distilbert-base-uncased | 1060 | 1228 |
| ELECTRA [8] | google/electra-small-discriminator | 2099 | 2409 |

Table 4: List of transformers text models.

| name | full model name | $|\mathcal{N}|$ | $|\mathcal{V}|$ |
|---|---|---|---|
| GPT2 [29] | gpt2 | 1889 | 2179 |
| GPT Neo [6] 125M | EleutherAI/gpt-neo-125m | 2088 | 2378 |
| GPT Neo 2.7B | EleutherAI/gpt-neo-2.7B | 5488 | 6238 |
| BLOOM [32] 560M | bigscience/bloom-560m | 3667 | 4167 |
| BLOOM 3B | bigscience/bloom-3b | 4567 | 5187 |
| OPT [43] 350M | facebook/opt-350m | 3971 | 4581 |
| OPT 6.7B | facebook/opt-6.7b | 5245 | 6059 |
| LLaMA [40] 7B | default config* | 8330 | 8722 |

Table 5: List of large language models. *For LLaMA, we constructed a model using default configuration (7B parameters) provided by `transformers.LlamaConfig`.

## D.6 Solution stability of FastSA and Checkmate LP

Since both FastSA and Checkmate LP are randomized algorithms, the recomputation plans depend on the seed. We executed FastSA and Checkmate LP for various memory budgets and see how smoothly the solution change. Figure 9 shows the time-space tradeoff curve for different memory budgets. Since Checkmate LP could not solve the instances with more than 2300 nodes, we compared the performance difference on relatively small models. For each model, we set memory budgets as $0.15, 0.20, \ldots, 0.95, 1.00$ and plotted the results. In most cases, FastSA found even better solutions than Checkmate LP. Although FastSA is a randomized heuristic, its performance is stable and not heavily dependent on random seeds. On the other hand, Checkmate LP's performance is not stable, mainly due to LP-related issues as discussed in Appendix A.1. For instance, in ViT (small), one of the solutions found by Checkmate LP had almost 10 times overhead. Although running Checkmate LP with different seeds or hyperparameters may still yield good solutions, it requires tremendous computational resources.

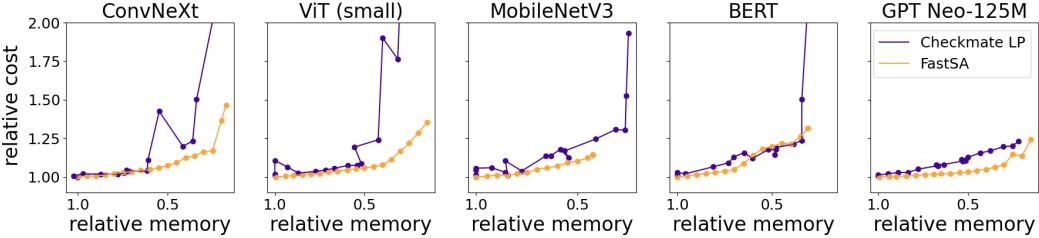

Figure 9: The time-space tradeoff curves for Checkmate LP and FastSA. For most settings, FastSA could find solutions with lower cost overhead than Checkmate LP for the same relative memory reduction. Checkmate LP is not stable due to LP relaxation. For MobileNetV3, FastSA could not find solutions for small memory budgets. This would be due to the limitation of the node grouping.

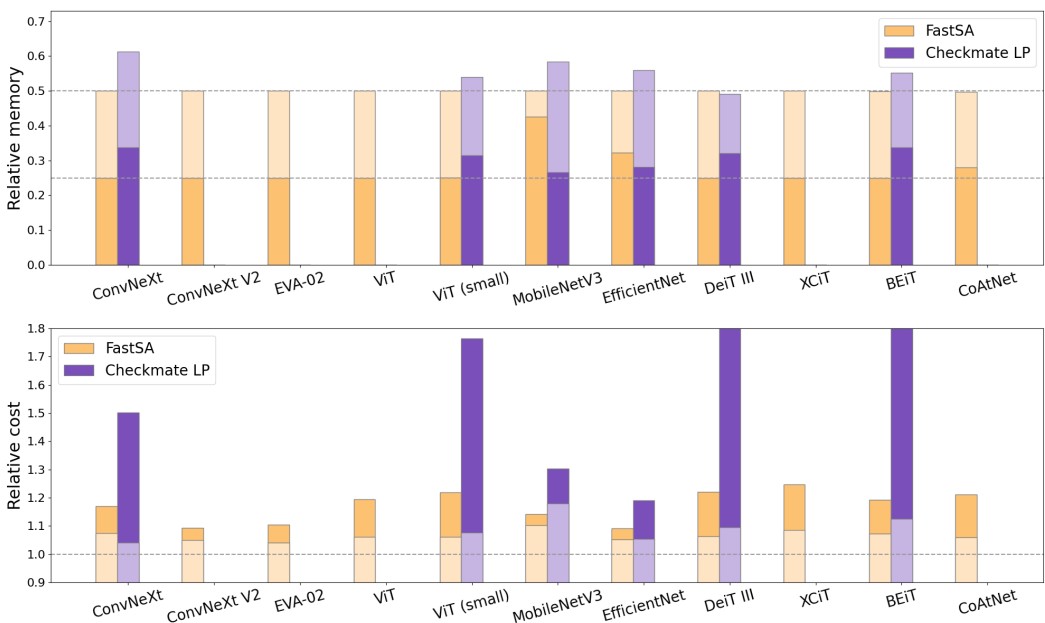

Figure 10: Time-space tradeoff for vision models with 25% (darker) and 50% (lighter) memory budgets. The batch size is 512. Checkmate LP was unable to find solutions within a 6-hour time limit for ViT and CoAtNet. The other models without Checkmate LP bars suffered from out-of-memory errors exceeding 100 GiB. The truncated results of the relative cost of Checkmate LP for DeiT III and BEiT were 2.82 and 3.52, respectively.

### D.7    Full experimental results for comparison with Checkmate LP

The graph in Figure 10 displays the recomputation results of our FastSA algorithm and Checkmate LP for all the vision models in Table 3. For ViT and CoAtNet, although Checkmate LP could run within a 100 GiB RAM, it failed to find a feasible solution within the 6-hour time limit. Furthermore, the recomputation overhead by Checkmate LP for DeiT III and BEiT was more than 2.8 times, possibly due to randomized rounding of non-binary LP solutions. While FastSA obtained better solutions in both time and memory for ViT (small), DeiT III, and BEiT, it could not reduce memory as much as Checkmate LP for MobileNetV3 and EfficientNet, mainly due to suboptimal node grouping.

Figure 11 displays the recomputation results for all the text models in Table 4 and recent language models in Table 5. For some text models, such as BERT and ELECTRA, our algorithm was unable to reduce memory usage sufficiently for a 25% budget, either due to limitations on the models

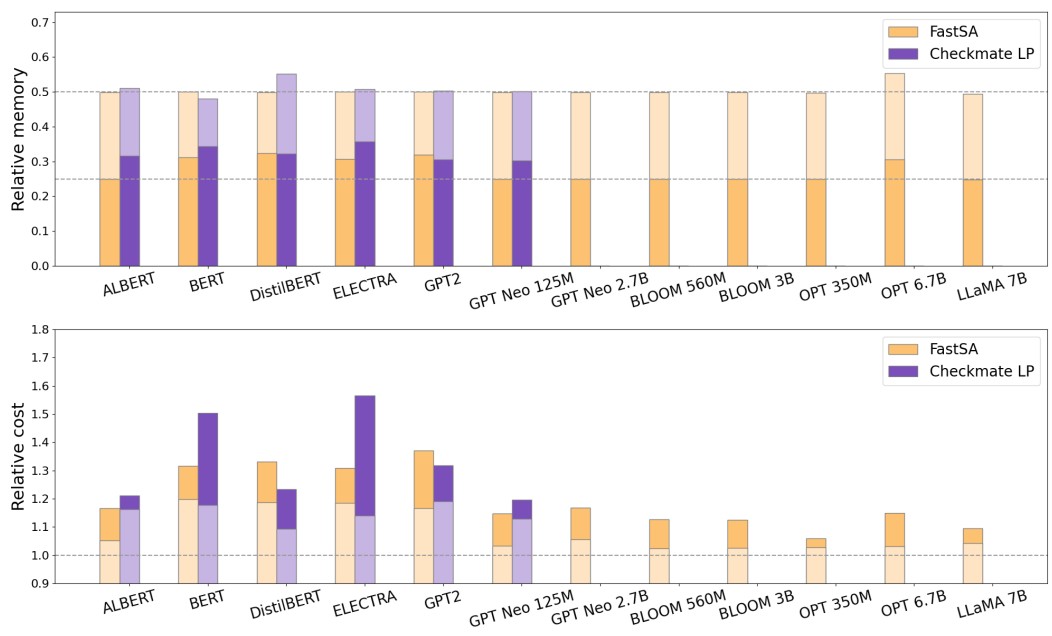

Figure 11: Time-space tradeoff for text models, including language models, with 25% (darker) and 50% (lighter) memory budgets. Checkmate LP failed for larger models due to out-of-memory errors exceeding 100 GiB.

themselves or convergence to suboptimal solutions. However, overall, our algorithm was successful in finding good recomputation plans with only a small increase in the recomputation overhead.

### D.8 Actual GPU memory reduction and time increase

Throughout the above experiments, we discussed the memory decrease and cost increase by estimated values. This is because it is hard to assign exact computational costs for each node without execution (note that our computational graphs were traced without execution) and/or values may share the same physical storage in actual GPU memory. In this section, we provide a summary of the results of actual memory usage and execution time measured on an NVIDIA A100 80GB GPU and discuss the differences between the actual metrics and the simulated ones.

Figure 12 displays the difference of simulated and actual CUDA memory usages for various models for three cases: no optimization, 50% budget, and 25% budget. On average, the actual GPU memory usages were less than the simulated values by 10% for no optimization cases, but could be up to 5% more for 25% budget cases. In most cases, the simulated memory usages were higher than the allocated usages, mainly because the memory simulation is not precise enough. Notably, operations like `reshape`, `expand`, and slicing may produce new views of the original tensor and do not allocate new GPU memory. Our memory simulation does not consider this information, leading to overestimation of the memory usage. Memory aliasing, tensor views, and node-intermediate memory usage can account for these differences. For the 25% budgets, the simulated and allocated memory usages are similar for all models. However, in some instances, the allocated memory exceeds the simulated memory, which may be due to an inaccurate memory simulation. One of the reasons is that we do not consider the node-intermediate memory usage for simulation, which underestimates the memory usage. We can address this issue in our problem setting by simulating this value (please refer to Appendix B.1).

Figure 13 shows the difference between the actual increases in execution time and the simulated ones by the unit cost. We observed that the simulated cost is relatively close to the actual measured overhead. The difference in simulated time increase from the actual time increase was at most 7% and 12% for 50% and 25% memory budgets respectively. We observe that practical recomputation plans can be successfully obtained for many models using the unit cost. However, the reason behind

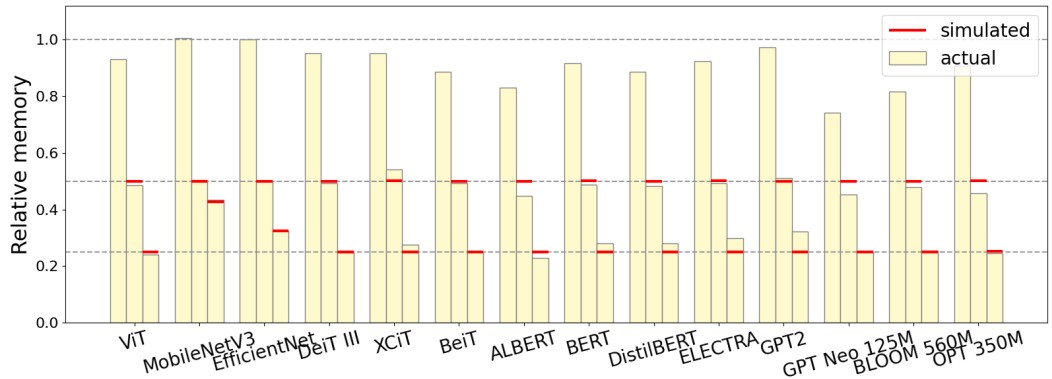

Figure 12: The actual memory usage of optimized models by FastSA. For each model, the three bars indicate the actual memory usage of the model when executed without optimization, and with 50% and 25% budget constraints. The simulated memory is indicated by the red line. The actual memory is the total size of CUDA tensor allocated by the PyTorch CUDA allocator. As detailed in Section D.5, the batchsizes and context lengths are modified to run the original model without recomputation for comparison.

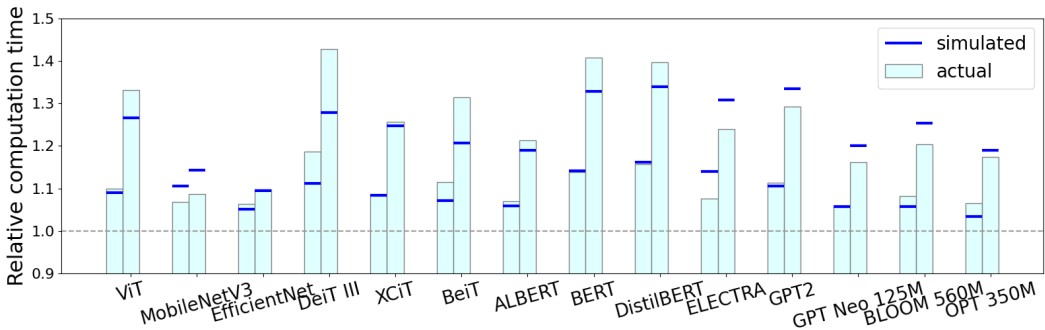

Figure 13: Overhead of execution time by recomputation. For each model, the two bars correspond to 50%, and 25% memory budget setting, respectively. The simulated increase of the computational cost is indicated by the blue line.

similar simulated and actual overhead is a balanced mix of operators with high costs, such as matrix multiplication, and operators with negligible costs, such as reshape, effectively averaging the cost. By accurately profiling the operators, it is possible to obtain more efficient recomputation plans.

