# OpenReview forum: "A fast heuristic to optimize time-space tradeoff for large models"
_NeurIPS.cc/2023/Conference — NeurIPS 2023 poster_

### Official Review · Reviewer_wXFY · 2023-06-29

**Soundness:** 2 fair
**Presentation:** 2 fair
**Contribution:** 3 good
**Rating:** 6
**Confidence:** 4

**Summary:**

This paper proposed an algorithm for the rematerialization problem. The proposed algorithm is based on simulated annealing.

**Strengths:**

Please see the questions section.

**Weaknesses:**

Please see the questions section.

**Questions:**

- As far as I know, Checkmate has two versions: 1) MILP, 2) Rounded LP. It looks like that the authors consider the rounded LP only. Based on my experience, rounding leads to very low quality solutions especially for complex DAGs. I understand because of gurobi being a commercial solver, it could be difficult for the authors to obtain a comparison for the gurobi version. One solution to this is to use the SCIP solver (maybe use directly its API or via CVXPY) or even use the available MILP solver in Google Or-tools. More importantly, what I want to get at is that it's not clear to me how close to the optimal the solution produced by the proposed algorithm is. This could be numerically studied by taking a smaller graph (maybe 100 nodes or so) and then solving the MILP of Checkmate using SCIP to global optimality. Then, the authors could report how close to this number their solution is. I expect this would make the results more complete.

In summary, I am convinced that the proposed method works better than rounded LP, but also want to point out that rounded LP doesn't work well anyways. I'd be curious to see how far it's from optimal at least for graphs of manageable size.

- Line 202: "Since the exact cost of a node cannot be determined before execution, we assumed that all operators have the same cost". Perhaps the authors consider profiling the operator durations? Or could it be possible to come up with a more sophisticated (than unit cost) way of assigning durations to the operators?

- I am not expecting the authors to be familiar with this paper or include it in their numerical comparisons since it came out very recently, but it's quite relevant: [r1] "Moccasin: Efficient Tensor Rematerialization for Neural Networks" from ICML 2023. The authors of [r1] show that there exists an nlogn-variable formulation (as opposed to the n^2-variable formulation of Checkmate).

- I think this claim in line 311 may need to be made more precise: "This complexity scales quadratically with the size of the input computational graph, making it a time and memory consuming process." The number of decision variables scales quadratically, not the overall algorithm complexity. The overall algorithm complexity is much worse than quadratic since it's a MILP.

- input(n), output(n): Maybe a more standard terminology is to use parents/children predecessors/successors?

- I apologize if this is already mentioned in the text but, what is the difference between "FastSA" and "FastSA Only" in Figure 4?

- How do you define the optimization time in Figure 4? This looks a bit confusing to me, since we know that by running a milp solver with infinite time will get us optimal.

**Limitations:**

Please see the questions section.

---

> ### Author Rebuttal · Authors · 2023-08-08
>
> We really appreciate your feedback emphasizing the importance of comparing FastSA results to Checkmate ILP. In our original paper, we relied solely on Checkmate LP due to licensing issues and the large sizes of the models. However, we now understand the necessity of a comparison with Checkmate ILP to evaluate FastSA's solution quality more effectively.
>
> As detailed in the global author rebuttal, we obtained a trial license of Gurobi and carried out additional experiments. Given Checkmate ILP's inability to find feasible solutions for the models used in our original experiments within a 3600-second time frame, we resorted to using vgg11 and resnet18. As per your suggestion, these models, which are composed of approximately ~100 nodes, were used to compute optimal recomputation plans.
>
> Table 2 in the attached PDF illustrates the optimality gap between FastSA and Checkmate ILP. For vgg11, FastSA's recomputation plan at most includes one additional node than the optimal plan identified by Checkmate. For resnet18, FastSA's plan at most contains 4 more nodes than the optimal Checkmate plan, in a computation graph originally consisting of 171 nodes.
>
> Additionally, Table 1 summarizes the experimental results, incorporating FastSA's results alongside those from the Moccasin paper. The quality of FastSA's solutions can be evaluated here in comparison with the optimal solutions derived by Checkmate or Moccasin. FastSA outperforms both in the case of random layered (RL) models, owing to the substantial flexibility offered by these models in terms of their topological order. FastSA can concurrently optimize this topological order whilst minimizing memory footprint, with almost no additional recomputation nodes.
>
> Although we had to significantly reduce the model size as depicted in Table 2 to obtain the exact solution through MILP with Checkmate, we anticipate that FastSA presents promising opportunities for producing superior solutions than Checkmate for complex and larger models due to the greater degree of flexibility in the model's topological order.
>
> **Q: Line 202: "Since the exact cost of a node cannot be determined before execution, we assumed that all operators have the same cost". Perhaps the authors consider profiling the operator durations? Or could it be possible to come up with a more sophisticated (than unit cost) way of assigning durations to the operators?**
>
> A: Indeed, by utilizing profiled runtime or flops estimation as the node cost, a more precise recomputation plan can be derived. However, there is a limitation when naively profiling a computation graph — the initial memory footprint of that graph should fit within the GPU memory. This becomes a challenge when recomputation is indispensable to fit the model within a single GPU. To negotiate such scenarios, an initial run of FastSA with a unit cost can be performed to decrease the memory usage, followed by profiling, and after that a more suitable recomputation plan can then be developed based on the profiled runtime.
>
> **Q: I am not expecting the authors to be familiar with this paper or include it in their numerical comparisons since it came out very recently, but it's quite relevant: [r1] "Moccasin: Efficient Tensor Rematerialization for Neural Networks" from ICML 2023. The authors of [r1] show that there exists an nlogn-variable formulation (as opposed to the n^2-variable formulation of Checkmate).**
>
> A: Please check the above reply and the global author rebuttal.
>
> **Q: I think this claim in line 311 may need to be made more precise: "This complexity scales quadratically with the size of the input computational graph, making it a time and memory consuming process." The number of decision variables scales quadratically, not the overall algorithm complexity. The overall algorithm complexity is much worse than quadratic since it's a MILP.**
>
> A: We apologize for the unclear explanation. We will revise it in our final version.
>
> **Q: input(n), output(n): Maybe a more standard terminology is to use parents/children predecessors/successors?**
>
> A: We will revise that section in our final version.
>
> **Q: I apologize if this is already mentioned in the text but, what is the difference between "FastSA" and "FastSA Only" in Figure 4?**
>
> A: “FastSA Only” means only FastSA could find a solution and Checkmate LP could not find it within the time limit of 6 hours. We will add the explanation to the caption.
>
> **Q: How do you define the optimization time in Figure 4? This looks a bit confusing to me, since we know that by running a milp solver with infinite time will get us optimal.**
>
> A: Optimization time in Figure 4 is not the total node costs of the optimized computation graph, but the time spent by FastSA/Checkmate algorithm to find a suitable recomputation plan.

---

> > ### Comment · Reviewer_wXFY · 2023-08-14
> >
> > Thanks for the responses to my questions. The rebuttal resolves most of my concerns. The additional numerical results raise my confidence in the proposed method. Especially the fact that the topological ordering is optimized (so, no need to stick with a highly sub-optimal topological ordering such as random) is an important contribution of this paper, in my opinion. I've raised my score to 6.

---

### Official Review · Reviewer_a8KW · 2023-07-05

**Soundness:** 3 good
**Presentation:** 3 good
**Contribution:** 2 fair
**Rating:** 6
**Confidence:** 4

**Summary:**

This paper proposed a new method for optimizing recomputation in neural network training. It formalizes the recomputation as a sequence of nodes that indicate the computation schedule. It then uses simulated annealing with segment tree to find a sequence that optimizes throughput with a certain memory budget. Experiments show significantly reduced computation overhead compared to the state-of-the-art optimizer checkmate in memory restricted cases. It also has a significant lower solving time than checkmate.

**Strengths:**

1. The paper is clean and easy to follow.
2. The formulation of the problem is clear.
3. The proposed method is novel for this problem.
4. The evaluation is instructive and convincing. The real implementation is aligned well with the simulator.

**Weaknesses:**

1. The contribution is incremental. It does not open a new problem or new angle, and the technique is common.

**Questions:**

1. Figure 3: Why is there no checkmate bar for AVE?
2. Figure 3: Why is there not much difference between FastSA and Checkmate for 50%?
3. Is it true that FastSA cannot beat Checkmate on transformers?

**Limitations:**

The linearization of the graph and the relaxation used in the algorithm makes the solution sub-optimal.

---

> ### Author Rebuttal · Authors · 2023-08-08
>
> Thank you for your review.
>
> Acknowledging that our problem formulation shares resemblances with existing work, it's pivotal to highlight that, based on our knowledge, our algorithm is the first algorithm able to perform recomputation in arbitrary computational graphs and can be applied to real-world models involving thousands of nodes making our solution truly innovative and unique in this area. Existing algorithms such as Checkmate were capable of handling prevalent models of their time, such as ResNet. However, with the evolution of model sizes and complexities, these methods fall short when it comes to handling larger models like modern Transformers, which are heavily constrained by the memory capacity of contemporary devices. So, in this sense, our research tackles a problem that's becoming increasingly significant, offering new solutions and greatly pushing the current boundaries of what's achievable.
>
> Although simulated annealing (SA) would be a common idea to solve optimization problems, we utilized segment trees within our SA to reduce the time complexity of re-evaluating the peak memory after the mutations on the computation graph from naive O(n) to O(log n). This technique is the core of our algorithm allowing it to process large graphs over thousands of nodes.
>
> **Q: Figure 3: Why is there no checkmate bar for AVE?**
>
> A: The absence of an AVE bar for Checkmate in the figures is due to Checkmate's inability to find a solution within the six-hour time limit for some instances. This detail is covered in line 252 of the manuscript, but we acknowledge that the explanation might not have been sufficiently clear. For better clarity, a direct explanation will be added in the figure captions.
>
> **Q: Figure 3: Why is there not much difference between FastSA and Checkmate for 50%?**
>
> A: The relative ease of recomputation at this level allows both algorithms to effectively identify good solutions. However, as the budget reduces to 25%, the number of feasible computation orders decreases and the recomputation plans will become more complex, introducing challenges for rematerialization algorithms. This point will be better elaborated in the revised manuscript for improved understanding.
>
> **Q: Is it true that FastSA cannot beat Checkmate on transformers?**
>
> A: When dealing with transformer models, FastSA tends to outperform Checkmate. This is supported by a comparative analysis in Figure 12 of the supplementary materials for transformer models. For the six transformer models in Figure 12 where both FastSA and Checkmate results are available, FastSA has on average 9.6% smaller memory and 4.1% faster than Checkmate.

---

> > ### Comment · Reviewer_a8KW · 2023-08-19
> >
> > Thanks for the clarification.
> > This paper [1] came out last month, which is later than the neurips submission but very relevant. Could you add a discussion comparing your work with it in the revised version?
> >
> > [1] Rockmate: an Efficient, Fast, Automatic and Generic Tool for Re-materialization in PyTorch

---

> > > ### Author Response · Authors · 2023-08-21
> > > **About comparison between FastSA and Rockmate**
> > >
> > > We appreciate the reference to Rockmate, of which we were not aware at the time of this paper's submission. We plan to discuss and compare the results in the revised version of our manuscript. In the meantime, we would like to briefly outline the key distinctions between Rockmate and FastSA.
> > >
> > > Rockmate adapts a "block sequence" approach for models to reduce the heavy ILP computation associated with Checkmate. This involves precomputing recomputation strategies within blocks for several memory budgets using Checkmate, followed by a dynamic programming optimization of the global memory, named Rotor [Beaumont et al. 2019].
> > >
> > > Rockmate's strength over Checkmate lies in its ability to significantly expedite optimization time if the model is a sequential model of identical layers, thanks to the precomputation with Checkmate required only for a single block. However, Rockmate's disadvantage is its restriction to sequential models, and it does not scale for general computation graphs. Particularly, the block partitioning described in Appendix A.4 of Rockmate's paper does not reduce memory sufficiently if the computation graph has significant topological freedom.
> > >
> > > Conversely, FastSA caters to any form of computation graphs, and its optimization time is less dependent on network structure, allowing for flexible model modification during the development's trial and error stages.
> > >
> > > Given Rockmate's focus on specific models like ResNet50 or GPT2, which are sequential and repeat the same layers, it wouldn't be fair to only compare FastSA to these models. However, we found FastSA to yield better solutions on ResNet50 (-80.6% memory, +10.7% time) compared to Rockmate (-65% memory, +23% time).
> > >
> > > We will describe more details about the experimental results in our final revision.

---

> > > > ### Comment · Reviewer_a8KW · 2023-08-21
> > > >
> > > > Thanks for your further clarifications. I have no further questions.

---

### Official Review · Reviewer_kH45 · 2023-07-05

**Soundness:** 3 good
**Presentation:** 3 good
**Contribution:** 3 good
**Rating:** 6
**Confidence:** 3

**Summary:**

This paper proposes the Fast Simulated Annealing Algorithm (FastSA), based on the Add-max segment tree and simulated annealing, to optimize memory usage and training time. Furthermore, FastSA introduces grouped nodes to aid the convergence of simulated annealing and effectively reduce the peak memory. It can also be extended to incorporate other memory optimization techniques such as offloading. As a result, FastSA successfully finds optimal solutions, even for large language models where existing checkmate algorithms fail, leading to optimized training speed and memory usage.

**Strengths:**

This paper presents a **novel approach** that differs from the existing checkmate method in three key aspects.
+ First, it formulates the problem as optimizing the sequence of nodes by defining each node to represent a single operator and introducing the concept of Lifetime.
+ Second, it proposes an algorithm that effectively combines the add-max segment tree and simulated annealing to solve this problem.
+ Third, it optimizes the process through grouping.

These ideas are not only highly innovative but also demonstrated to **successfully optimize memory and training time when applied to large models**. This highlights the significance of the findings in this paper.


**Weaknesses:**

**Lack of comparisions**

The FastSA finds the optimal solution based on the concept of lifetime. This approach shares many similarities with the recently proposed event-based optimization algorithm called Moccasin [1]. Specifically, Moccasin also claims to achieve superior performance compared to MLIP-based Checkmate by utilizing time interval information for optimization. However, the FastSA paper does not mention this and lacks quantitative or qualitative comparisons.

[1] Moccasin: Efficient Tensor Rematerialization for Neural Networks

**Questions:**

If a proper comparison between this algorithm and Moccasin is conducted and it is demonstrated that FastSA achieves significantly better performance, I am willing to raise the score to 7 or higher.

+ A qualitative comparison between the Moccasin algorithm and FastSA, highlighting the superior aspects of FastSA.

+ A quantitative comparison between the Moccasin algorithm and FastSA based on experimental results.



**Limitations:**

The authors provide a thorough discussion of the limitations and potential future work in the paper.

---

> ### Author Rebuttal · Authors · 2023-08-09
>
> Thank you for your review.
>
> We acknowledge and appreciate the reference to Moccasin, which was unavailable at the time of our paper's submission, but is indeed an important comparison to make. Consequently, we have conducted further comparisons between Moccasin and FastSA, which we will include, along with a detailed explanation, in our revised manuscript.
>
> **Q: A qualitative comparison between the Moccasin algorithm and FastSA, highlighting the superior aspects of FastSA.**
>
> A: Different from Checkmate and FastSA, Moccasin uses Constraint Programming (CP) to address the rematerialization problem. By focusing on lifetime intervals, Moccasin reduced the number of integer variables in CP to linear size to the number of nodes, compared to the quadratic number of boolean variables in Checkmate. However, the recomputation plans produced by Moccasin may not be efficient enough, because its search space for the solutions is $O(2^{Cn} + n^{Cn})$ and is significantly narrower than Checkmate’s $O(2^{n^2 + nm})$ or FastSA’s, where C is the hyperparameter representing the maximum number of recomputations for each node and in the open-source implementation, it is configured to C=2. The solution space of FastSA is even larger than Checkmate, meaning that FastSA can find even better recomputation plans than both Checkmate or Moccasin. Setting C=2 is practical in terms of execution time, but then the memory reduction on sequential models is limited to O(√n), whereas it is feasible to achieve O(log n) or O(1) memory by allowing more than two recomputations for each node.
>
> In fact, the experiments in the original paper of Moccasin (also cited in Table 1 of our attached pdf file), were conducted for 90% and 80% memory budgets, where the peak memory can be reduced with a small number of recomputations. However, as shown in our original experiments, we can even reduce memory to 25% given modern large-scale models. In such situations, the constraint of C=2 in Moccasin can significantly limit the solution quality. By setting a larger C value, the space of candidate solutions increases exponentially, limiting the potential applicability of aggressive memory reductions. For more explanation on the size of search space, please refer to Table 1 of Moccasin paper.
>
> **Q: A quantitative comparison between the Moccasin algorithm and FastSA based on experimental results.**
>
> A: As shown in Table 1 of our attached pdf, FastSA consistently identified solutions with a smaller Total Duration Increase (TDI), which refers to the recomputation overhead, faster than Moccasin in many instances. Notably, for the random layered (RL) case, FastSA successfully minimized memory consumption up to 4.9% smaller TDI than Moccasin without the need for additional recomputation nodes, simply through optimizing the topological ordering. While Moccasin operates faster than Checkmate, it takes approximately an hour for cases nearing 1000 nodes, whereas FastSA can find superior solutions in a fraction of this time.
>
> It's important to note that constraint optimization problems, as addressed by Moccasin and Checkmate, generally demand a substantial computational time to arrive at feasible solutions, more so when the solution space shrinks. As a result, even with equal node numbers, the computational time escalates considerably as the budget tightens. This is evident in Table 2 of our additional materials, which illustrates an example of Checkmate results. Checkmate took more than 40x longer optimization time for resnet18 when the budget was tightened from 80% to 60%. Accordingly, the benefits of employing FastSA are even more pronounced when navigating tighter budgets as the optimization time only increased up to 5% when the budget was changed from 90% to 50% in the case of resnet18.
>
> We trust that we have adequately addressed your concerns. Thank you once more for your thoughtful review.

---

> > ### Comment · Reviewer_kH45 · 2023-08-17
> > **Response to Authors**
> >
> > I sincerely appreciate the author's effort and response. Thanks to this, I now clearly understand what advantages FastSA has over other algorithms like Moccasin.

---

### Official Review · Reviewer_w4oG · 2023-07-11

**Soundness:** 3 good
**Presentation:** 2 fair
**Contribution:** 3 good
**Rating:** 6
**Confidence:** 3

**Summary:**

This paper introduces a fast simulating annealing combined heuristics approach for gradient checkpoint/recomputation.
The solution achieves a significant memory reduction of 73% with an average recomputation cost of 18%.
It outperforms the state-of-the-art MILP-based technique Checkmate in terms of runtime by orders of magnitude, while still meeting the memory budget requirements.

Unlike Checkmate, which utilizes MILP optimization on the graph, this work employs a sequence of nodes to represent values and a binary variable LS(v, t) -> {0, 1} to indicate if a value v is in memory at time t. The optimization objective is defined as the maximum of the resource utilization function M and the budgeted compute cost C. To optimize the solution, the paper utilizes an add-max segment tree structure for efficient interval evaluation. The optimization algorithm employed is simulating annealing (SA). The approach allows for three mutations in the sequence: add, remove, and rotate, enabling effective exploration of different configurations.

**Strengths:**

1. The approach presented in this work introduces a novel and scalable solution for addressing the recomputation problem. Rather than formulating the problem as MILP which can be computationally expensive, this work reformulates it using sequences of nodes and utilizes an efficient data structure to evaluate its objective function. The problem is then optimized using the well-known simulated annealing algorithm, eliminating the need for a commercial solver.

2. By adopting this approach, this work manages to reduce the memory requirements to meet the specified 25% and 50% memory budget without incurring high computation costs.

3. The optimization time of this approach is notably faster, around 3-4 orders of magnitude compared to the previous state-of-the-art technique Checkmate.


**Weaknesses:**

1. The baseline Checkmate is run using the open-source OR-tool instead of Gurobi. It not only affects the runtime (which can be orders of magnitude longer) but also the quality of the results of MILP.

2. The comparison of end-to-end training time between the proposed approach and Checkmate is not provided. The increase in compute cost may not necessarily lead to improved performance if the workload is not compute-bound.

3. The topic could probably be better evaluated and discussed at a systems conference.


**Questions:**

1. It would be important to re-evaluate the Checkmate results using Gurobi.

2. Can you show a training time comparison of the solutions generated by SA and Checkmate?

3. Can you provide some explanation of why Checkmate as an ILP failed to meet the memory budget requirements? Are the constraints not precisely formulated in Checkmate?


**Limitations:**

- The grouping heuristics for nodes might not be optimal
- There are many other GPU memory usage factors to consider in the optimization.

---

> ### Author Rebuttal · Authors · 2023-08-09
>
> Thank you for your review. In our original experiments, we compared our algorithm to Checkmate LP, instead of the exact ILP, using open-source LP solver OR-Tools due to the prohibitive cost of commercial ILP/LP solvers. However, we acknowledge that the evaluation of the solution quality of our algorithm is important and we procured a trial license of Gurobi and conducted an initial exploratory study. Due to the short rebuttal period, we did not replace all of the Checkmate LP results solved by an OSS solver with Gurobi. Instead, we concentrated on verifying the optimality of FastSA’s solutions by solving Checkmate ILP (no relaxation) with Gurobi. We will replace the results in the next version of the manuscript.
>
> Also as summarized in the global author rebuttal, we first tried to evaluate all of the models used in our original experiments by Checkmate ILP with Gurobi, but even for smaller models within 1000 nodes, none of them was solved within the time limit of 3600 seconds. Therefore, we prepared much smaller instances (vgg11, resnet18) for calculating optimal solutions.
>
> Table 2 in the attached pdf shows the optimality gap between FastSA and Checkmate ILP. For vgg11, FastSA’s recomputation plan contained at most one more node than Checkmate optimal. For resnet18, FastSA’s plan contained at most 4 more nodes than Checkmate optional, where the original computation graph has 171 nodes. In addition, Table 1 summarizes the experiment results by adding FastSA's results to the table in the paper of Moccasin, a recently published recomputation algorithm at ICML 2023. Here, we can also discuss the solution quality of FastSA, compared to Checkmate/Moccasin optimals. In the case of RL (random layered), FastSA's results are better than Checkmate's or Moccasin's. This is because there is a high degree of freedom in model topological order in the RL cases, and FastSA can simultaneously optimize the topological order while minimizing the memory footprint with almost no additional recomputation nodes.
>
> **Q: It would be important to re-evaluate the Checkmate results using Gurobi.**
>
> A: Please kindly refer to the above comments and the general response for more details on this.
>
> **Q: Can you show a training time comparison of the solutions generated by SA and Checkmate?**
>
> A: In our paper, we did not directly assess the end-to-end training time. Instead, akin to prior work on recomputation [Kumar et al. NeurIPS 2019, Kusumoto et al. NeurIPS 2019, Jain et al 2020, Bartan et al. 2023], we focused on quantifying the additional computational time required when the memory budget for the model is constrained. However, an estimation of end-to-end training time can be inferred from our optimization results. Generally, the memory used in training directly corresponds with the batch size. Hence, if the memory is reduced to 1/k, it enables handling a batch size that is roughly k times larger. Assuming an average increase in computational time per operator, symbolized as c (a value estimable through benchmarks), the throughput of end-to-end training is likely to improve by a factor of k/c.
>
> For distributed training, data transfer bandwidth can pose a bottleneck rather than the arithmetic computational cost, rendering the end-to-end experimental setup more complex. However, as detailed in Appendix B.3, FastSA can be adapted to problem settings that include data transfer. With a suitably defined objective function, it may potentially accommodate these scenarios in the future.
>
> **Q: Can you provide some explanation of why Checkmate as an ILP failed to meet the memory budget requirements? Are the constraints not precisely formulated in Checkmate?**
>
> A: The experiment results of Checkmate in our paper are all solutions from the LP relaxation problem setting, not the ILP. Since SCIP, an open-source ILP solver, could not solve any instance in our original experiments within the time limit of 6 hours, we decided to solve the relaxed problem. The solution obtained from LP relaxation is generally not an integer, and the memory budget constraint may be violated during the randomized rounding process. More detailed explanations about this can be found in Section 5.2 of the original paper of Checkmate [Jain et al. 2020].
>
> As detailed previously, to evaluate the solution quality compared to the ILP optimal, we added extra results on vgg11 and resnet18 in the attached pdf file. Please note that even with Gurobi, we could not find optimal solutions within 3600s even for the smallest models used in our original experiments.
>
> We hope this adequately addresses your concerns and appreciate your feedback.

---

> > ### Comment · Reviewer_w4oG · 2023-08-15
> >
> > Thanks for the responses to the questions. The rebuttal has addressed the majority of my concerns. I raised my score from 5 to 6.

---

### Author Rebuttal · Authors · 2023-08-08

Dear reviewers

We appreciate the insightful feedback. The review comments have been proven to be very valuable and help to increase the quality of the paper. Here, we provide additional results regarding two common concerns; (1) Comparison with Moccasin [Bartan et al.], a rematerialization algorithm recently published in ICML 2023, and (2) The solution quality of FastSA (our algorithm), compared to Checkmate when solved with the exact ILP (integer linear programming), using a strong commercial solver, Gurobi.
For other questions, please refer to the individual replies.

**Q: Comparison of FastSA and Moccasin [Bartan et al. ICML 2023]**

A: We have benchmarked FastSA against Moccasin using the publicly available data that the authors released and observed that FastSA is ~100 times faster and could find solutions with less or equal recomputation overhead than Moccasin for 10/12 cases when evaluated in graphs with ~1000 nodes.

Moccasin formulates the problem of rematerialization by constraint programming (CP), which is similar to Checkmate’s ILP. Compared to Checkmate, Moccasin reduces the number of boolean variables from quadratic to linear, resulting in faster execution.
However, the scope of feasible solutions within Moccasin is significantly limited owing to the role played by the C hyperparameter, which represents the maximum number of times a value can be rematerialized. In practice, C is set to 2 because the search space of Moccasin is $O(2^{Cn} + n^{Cn})$ and thus its computational time grows exponentially for larger C. Specifically when dealing with sequential models, this setting of C=2 imposes a limit on the memory reduction to O(√n), despite the existence of recomputation sequences with O(log n) or O(1) memory utilization that can be achieved by both FastSA and Checkmate.

Table 1 in the attached pdf file shows the comparison among Checkmate, Moccasin, and FastSA. For RL (random layered) graphs, FastSA is able to find better solutions than both Checkmate and Moccasin. Especially, for 90% budget cases such as RL1 and RL2, FastSA reduced the memory budget up to 22.6% just by optimizing topological ordering without adding extra recomputation nodes.

**Q: How close to the optimal the solution produced by FastSA is?**

A: In the original version of the FastSA paper, a comparison with Checkmate ILP was not performed due to the prohibitive cost of commercial ILP solver licenses. Additionally, open-source solvers such as SCIP failed to complete any of the experiments within a reasonable time frame. However, as pointed out by reviewers wXFY and w4oG, a rigorous comparison with an ILP solver is necessary to ensure fairness.
To accommodate this, we procured a trial Gurobi license and reiterated all experiments on relatively small models (within 1000 nodes, 50% budget) in the evaluation section of the paper after cross-verifying our Checkmate ILP+Gurobi implementation with the Checkmate paper experiments. Regrettably, even with Gurobi, the experiments could not be completed within a time limit of 3600 seconds due to the considerable size of the models involved. As per reviewer wXFY's suggestion, we have incorporated graphs with approximately ~100 nodes such as vgg11 or resnet18 and obtained their optimal solutions using Checkmate ILP+Gurobi.

Table 2 in the attached PDF presents a comparison between Checkmate and FastSA for smaller instances. For vgg11, under the tightest budget constraint (80%), FastSA identified the same recomputation plan as Checkmate. For resnet18, wherein the tightest budget that allowed an optimal solution was 60%, FastSA's Total Duration Increase (TDI), which means recomputation overhead, was 1.2% above Checkmate's TDI. Even though FastSA did not determine an optimal recomputation plan for this scenario, its runtime is almost 100 times faster than Checkmate's, suggesting its potential for integration into neural network compilers to optimize a diverse range of large models

Table 1 also compares Checkmate ILP and FastSA. For the RL cases, FastSA found solutions with a maximum TDI of 0.3%, whereas all Checkmate solutions had a maximum TDI of 0.8%. In these scenarios, FastSA's solutions superseded those of Checkmate as FastSA was successful in reducing the peak memory through optimizing the topological ordering. Please refer to Appendix A.1 for a more detailed explanation of this constraint in Checkmate when dealing with computation graphs that offer high topological flexibility.

---

### Decision · Program_Chairs · 2023-09-21

**Decision:**

Accept (poster)

**Comment:**

The paper proposes a new method to optimize recomputation during neural network training (recomputation = a sequence of nodes in some computation graph that indicate the computation schedule during training). It's based on simulated annealing strategies that attempt to find a sequence that optimizes throughput with a certain memory budget. The paper boasts memory reductions by 73% with mild (18%) computational overheads on average.

The reviews were largely positive. The main concerns raised were that: 1) similar techniques have appeared in the related literature; 2) the topic might be more appropriate for a MLSys style venue.

On balance, the method shows enough improvements, and there's enough interest in NeurIPS in training big models, that including it in the program makes sense.